# END-TO-END EGOSPHERIC SPATIAL MEMORY

**Daniel Lenton** [1]**, Stephen James** [1]**, Ronald Clark** [2]**, Andrew J. Davison** [1]
[1] Dyson Robotics Lab, [2] Department of Computing, Imperial College London
`{djl11,slj12,ronald.clark,a.davison}@ic.ac.uk`

## ABSTRACT

Spatial memory, or the ability to remember and recall specific locations and objects, is central to autonomous agents' ability to carry out tasks in real environments. However, most existing artificial memory modules are not very adept at storing spatial information. We propose a parameter-free module, Egospheric Spatial Memory (ESM), which encodes the memory in an ego-sphere around the agent, enabling expressive 3D representations. ESM can be trained end-to-end via either imitation or reinforcement learning, and improves both training efficiency and final performance against other memory baselines on both drone and manipulator visuomotor control tasks. The explicit egocentric geometry also enables us to seamlessly combine the learned controller with other non-learned modalities, such as local obstacle avoidance. We further show applications to semantic segmentation on the ScanNet dataset, where ESM naturally combines image-level and map-level inference modalities. Through our broad set of experiments, we show that ESM provides a general computation graph for embodied spatial reasoning, and the module forms a bridge between real-time mapping systems and differentiable memory architectures. Implementation at: `https://github.com/ivy-dl/memory`.

## 1 INTRODUCTION

Egocentric spatial memory is central to our understanding of spatial reasoning in biology (Klatzky, 1998; Burgess, 2006), where an embodied agent constantly carries with it a local map of its surrounding geometry. Such representations have particular significance for action selection and motor control (Hinman et al., 2019). For robotics and embodied AI, the benefits of a persistent local spatial memory are also clear. Such a system has the potential to run for long periods, and bypass both the memory and runtime complexities of large scale world-centric mapping. Peters et al. (2001) propose an EgoSphere as being a particularly suitable representation for robotics, and more recent works have utilized ego-centric formulations for planar robot mapping (Fankhauser et al., 2014), drone obstacle avoidance (Fragoso et al., 2018) and mono-to-depth (Liu et al., 2019).

In parallel with these ego-centric mapping systems, a new paradigm of differentiable memory architectures has arisen, where a memory bank is augmented to a neural network, which can then learn read and write operations (Weston et al., 2014; Graves et al., 2014; Sukhbaatar et al., 2015). When compared to Recurrent Neural Networks (RNNs), the persistent memory circumvents issues of vanishing or exploding gradients, enabling solutions to long-horizon tasks. These have also been applied to visuomotor control and navigation tasks (Wayne et al., 2018), surpassing baselines such as the ubiquitous Long Short-Term Memory (LSTM) (Hochreiter & Schmidhuber, 1997).

We focus on the intersection of these two branches of research, and propose Egospheric Spatial Memory (ESM), a parameter-free module which encodes geometric and semantic information about the scene in an ego-sphere around the agent. To the best of our knowledge, ESM is the first end-to-end trainable egocentric memory with a full panoramic representation, enabling direct encoding of the surrounding scene in a 2.5D image.

We also show that by propagating gradients through the ESM computation graph we can learn features to be stored in the memory. We demonstrate the superiority of learning features through the ESM module on both target shape reaching and object segmentation tasks. For other visuomotor control tasks, we show that even without learning features through the module, and instead directly projecting image color values into memory, ESM consistently outperforms other memory baselines.

Through these experiments, we show that the applications of our parameter-free ESM module are widespread, where it can either be dropped into existing pipelines as a non-learned module, or end-to-end trained in a larger computation graph, depending on the task requirements.

## 2 RELATED WORK

### 2.1 MAPPING

Geometric mapping is a mature field, with many solutions available for constructing high quality maps. Such systems typically maintain an allocentric map, either by projecting points into a global world co-ordinate system (Newcombe et al., 2011; Whelan et al., 2015), or by maintaining a certain number of keyframes in the trajectory history (Zhou et al., 2018; Bloesch et al., 2018). If these systems are to be applied to life-long embodied AI, then strategies are required to effectively select the parts of the map which are useful, and discard the rest from memory (Cadena et al., 2016).

For robotics applications, prioritizing geometry in the immediate vicinity is a sensible prior. Rather than taking a world-view to map construction, such systems often formulate the mapping problem in a purely ego-centric manner, performing continual re-projection to the newest frame and pose with fixed-sized storage. Unlike allocentric formulations, the memory indexing is then fully coupled to the agent pose, resulting in an ordered representation particularly well suited for downstream egocentric tasks, such as action selection. Peters et al. (2001) outline an EgoSphere memory structure as being suitable for humanoid robotics, with indexing via polar and azimuthal angles. Fankhauser et al. (2014) use ego-centric height maps, and demonstrate on a quadrupedal robot walking over obstacles. Cigla et al. (2017) use per-pixel depth Gaussian Mixture Models (GMMs) to maintain an ego-cylinder of belief around a drone, with applications to collision avoidance (Fragoso et al., 2018). In a different application, Liu et al. (2019) learn to predict depth images from a sequence of RGB images, again using ego reprojections. These systems are all designed to represent only at the level of depth and RGB features. For mapping more expressive implicit features via end-to-end training, a fully differentiable long-horizon computation graph is required. Any computation graph which satisfies this requirement is generally referred to as memory in the neural network literature.

### 2.2 MEMORY

The concept of memory in neural networks is deeply coupled with recurrence. Naive recurrent networks have vanishing and exploding gradient problems (Hochreiter, 1998), which LSTMs (Hochreiter & Schmidhuber, 1997) and Gated Recurrent Units (GRUs) (Cho et al., 2014) mediate using additive gated structures. More recently, dedicated differentiable memory blocks have become a popular alternative. Weston et al. (2014) applied Memory Networks (MemNN) to question answering, using hard read-writes and separate training of components. Graves et al. (2014) and Sukhbaatar et al. (2015) instead made the read and writes 'soft' with the proposal of Neural Turing Machines (NTM) and End-to-End Memory Networks (MemN2N) respectively, enabling joint training with the controller. Other works have since conditioned dynamic memory on images, for tasks such as visual question answering (Xiong et al., 2016) and object segmentation (Oh et al., 2019). Another distinct but closely related approach is self attention (Vaswani et al., 2017). These approaches also use key-based content retrieval, but do so on a history of previous observations with adjacent connectivity. Despite the lack of geometric inductive bias, recent results demonstrate the amenability of general memory (Wayne et al., 2018) and attention (Parisotto et al., 2019) to visuomotor control and navigation tasks.

Other authors have explored the intersection of network memory and spatial mapping for navigation, but have generally been limited to 2D aerial-view maps, focusing on planar navigation tasks. Gupta et al. (2017) used an implicit ego-centric memory which was updated with warping and confidence maps for discrete action navigation problems. Parisotto & Salakhutdinov (2017) proposed a similar setup, but used dedicated learned read and write operations for updates, and tested on simulated Doom environments. Without consideration for action selection, Henriques & Vedaldi (2018) proposed a similar system, but instead used an allocentric formulation, and tested on free-form trajectories of real images. Zhang et al. (2018) also propose a similar system, but with the inclusion of loop closure. Our memory instead focuses on local perception, with the ability to represent detailed 3D geometry in all directions around the agent. The benefits of our module are complementary to existing 2D methods, which instead focus on occlusion-aware planar understanding suitable for navigation.

## 3 METHOD

In this section, we describe our main contribution, the egospheric spatial memory (ESM) module, shown in Figure 1. The module operates as an Extended Kalman Filter (EKF), with an egosphere image $\mu_t \in \mathbb{R}^{h_s \times w_s \times (2+1+n)}$ and its diagonal covariance $\Sigma_t \in \mathbb{R}^{h_s \times w_s \times (1+n)}$ representing the state. The egosphere image consists of 2 channels for the polar and azimuthal angles, 1 for radial depth, and $n$ for encoded features. The angles are not included in the covariance, as their values are implicit in the egosphere image pixel indices. The covariance only represents the uncertainty in depth and features at these fixed equidistant indices, and diagonal covariance is assumed due to the large state size of the images. Image measurements are assumed to come from projective depth cameras, which similarly store 1 channel for depth and $n$ for encoded features. We also assume incremental agent pose measurements $u_t \in \mathbb{R}^6$ with covariance $\Sigma_{u_t} \in \mathbb{R}^{6 \times 6}$ are available, in the form of a translation and rotation vector. The algorithm overview is presented in Algorithm 1.

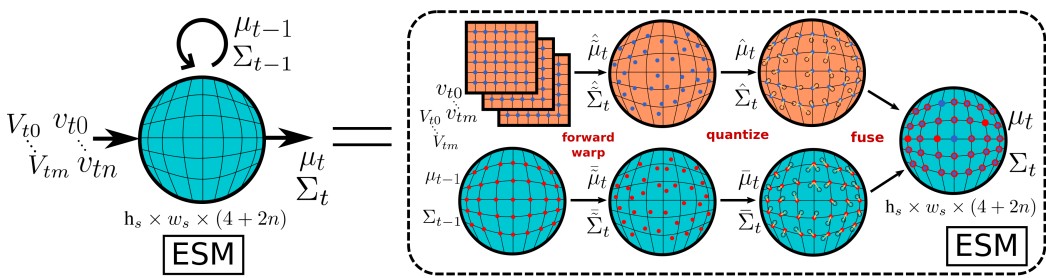

Figure 1: Overview of the ESM module. The module consists of projection and quantization steps, used to bring the belief from the previous agent frame to the current agent frame.

First, the motion step takes the state from the previous frame $\mu_{t-1}, \Sigma_{t-1}$ and transforms this into a predicted state for the current frame $\bar{\mu}_t, \bar{\Sigma}_t$ via functions $f_m, F_m$ and the incremental pose measurement $u_t$ with covariance $\Sigma_{u_t}$. Then in the observation step, we use measured visual features $(v_{ti} \in \mathbb{R}^{h_{vi} \times w_{vi} \times (1+n)})_{i \in \{1,\dots,m\}}$ with diagonal covariances $(V_{ti} \in \mathbb{R}^{h_{vi} \times w_{vi} \times (1+n)})_{i \in \{1,\dots,m\}}$ originated from $m$ arbitrary vision sensors, and associated pose measurements $(p_{ti} \in \mathbb{R}^6)_{i \in \{1,\dots,m\}}$ with covariances $(P_{ti} \in \mathbb{R}^{6 \times 6})_{i \in \{1,\dots,m\}}$, to produce a new observation of the state $\hat{\mu}_t \in \mathbb{R}^{h_s \times w_s \times (2+1+n)}$, again with diagonal covariance $\hat{\Sigma}_t \in \mathbb{R}^{h_s \times w_s \times (1+n)}$, via functions $f_o$ and $F_o$. The measured poses also take the form of a translation and rotation vector.

**Algorithm 1:** ESM Step

1. Given: $f_m, F_m, f_o, F_o$
2. $\bar{\mu}_t = f_m(u_t, \mu_{t-1})$
3. $\bar{\Sigma}_t = F_m(u_t, \mu_{t-1}, \Sigma_{t-1}, \Sigma_{u_t})$
4. $\hat{\mu}_t = f_o((v_{ti}, p_{ti})_{i \in I})$
5. $\hat{\Sigma}_t = F_o((v_{ti}, p_{ti}, V_{ti}, P_{ti})_{i \in I})$
6. $K_t = \bar{\Sigma}_t \left[ \bar{\Sigma}_t + \hat{\Sigma}_t \right]^{-1}$
7. $\mu_t = \bar{\mu}_t + K_t[\hat{\mu}_t - \bar{\mu}_t]$
8. $\Sigma_t = [I - K_t] \bar{\Sigma}_t$
9. return $\mu_t, \Sigma_t$

Finally, the update step takes our state prediction $\bar{\mu}_t, \bar{\Sigma}_t$ and state observation $\hat{\mu}_t, \hat{\Sigma}_t$, and fuses them to produce our new state belief $\mu_t, \Sigma_t$. We spend the remainder of this section explaining the form of the constituent functions. All functions in Algorithm 1 involve re-projections across different image frames, using forward warping. Functions $f_m, F_m, f_o$ and $F_o$ are therefore all built using the same core functions. While the re-projections could be solved using a typical rendering pipeline of mesh construction followed by rasterization, we instead choose a simpler approach and directly quantize the pixel projections with variance-based image smoothing to fill in quantization holes. An overview of the projection and quantization operations for a single ESM update step is shown in Fig. 1.

### 3.1 FORWARD WARPING

Forward warping projects ordered equidistant homogeneous pixel co-ordinates $pc_{f1}$ from frame $f_1$ to non-ordered non-equidistant homogeneous pixel co-ordinates $\tilde{pc}_{f2}$ in frame $f_2$. We use $\tilde{\mu}_{f2} = \{\tilde{\phi}_{f2}, \tilde{\theta}_{f2}, \tilde{d}_{f2}, \tilde{e}_{f2}\}$ to denote the loss of ordering following projection from $\mu_{f1} = \{\phi_{f1}, \theta_{f2}, d_{f1}, e_{f2}\}$, where $\phi, \theta, d$ and $e$ represent polar angles, azimuthal angles, depth and encoded features respectively. We only consider warping from projective to omni cameras, which corresponds

to functions $f_o, F_o$, but the omni-to-omni case as in $f_m, F_m$ is identical except with the inclusion of another polar co-ordinate transformation.

The encoded features are assumed constant during projection $\tilde{e}_{f2} = e_{f1}$. For depth, we must transform the values to the new frame in polar co-ordinates, which is a composition of a linear transformation and non-linear polar conversion. Using the camera intrinsic matrix $K_1$, the full projection is composed of a scalar multiplication with homogeneous pixel co-ordinates $pc_{f1}$, transformation by camera inverse matrix $K_1^{-1}$ and frame-to-frame $T_{12}$ matrices, and polar conversion $f_p$:

$$\{\tilde{\phi}_{f2}, \tilde{\theta}_{f2}, \tilde{d}_{f2}\} = f_p(T_{12}K_1^{-1}[pc_{f1} \odot d_{f1}]) \tag{1}$$

Combined, this provides us with both the forward warped image $\tilde{\mu}_{f2} = \{\tilde{\phi}_{f2}, \tilde{\theta}_{f2}, \tilde{d}_{f2}, \tilde{e}_{f2}\}$, and the newly projected homogeneous pixel co-ordinates $\tilde{pc}_{f2} = \{k_{ppr}\tilde{\phi}_{f2}, k_{ppr}\tilde{\theta}_{f2}, \mathbf{1}\}$, where $k_{ppr}$ denotes the pixels-per-radian resolution constant. The variances are also projected using the full analytic Jacobians, which are efficiently implemented as tensor operations, avoiding costly autograd usage.

$$\hat{\tilde{\Sigma}}_2 = J_V V_1 J_V^T + J_P P_{12} J_P^T \tag{2}$$

## 3.2 Quantization, Fusion and Smoothing

Following projection, we first quantize the floating point pixel coordinates $\tilde{pc}_{f2}$ into integer pixel co-ordinates $pc_{f2}$. This in general leads to quantization holes and duplicates. The duplicates are handled with a variance conditioned depth buffer, such that the closest projected depth is used, provided that it's variance is lower than a set threshold. This in general prevents highly uncertain close depth values from overwriting highly certain far values. We then perform per pixel fusion based on lines 6 and 7 in Algorithm 1 provided the depths fall within a set relative threshold, otherwise the minimum depth with sufficiently low variance is taken. This again acts as a depth buffer.

Finally, we perform variance based image smoothing, whereby we treat each $N \times N$ image patch $(\mu_{k,l})_{k \in \{1,..,N\}, l \in \{1,..,N\}}$ as a collection of independent measurements of the central pixel, and combine their variance values based on central limit theory, resulting in smoothed values for each pixel in the image $\mu_{i,j}$. Although we use this to update the mean belief, we do not smooth the variance values, meaning projection holes remain at prior variance. This prevents the smoothing from distorting our belief during subsequent projections, and makes the smoothing inherently local to the current frame only. The smoothing formula is as follows, with variance here denoted as $\sigma^2$:

$$\mu_{i,j} = \frac{\sum_k \sum_l \mu_{k,l} \cdot \sigma_{k,l}^{-2}}{\sum_k \sum_l \sigma_{k,l}^{-2}} \tag{3}$$

Given that the quantization is a discrete operation, we cannot compute it's analytic jacobian for uncertainty propagation. We therefore approximate the added quantization uncertainty using the numerical pixel gradients of the newly smoothed image $G_{i,j}$, and assume additive noise proportional to the $x$ and $y$ quantization distances $\Delta pc_{i,j}$:

$$\Sigma_{i,j} = \tilde{\Sigma}_{i,j} + G_{i,j}\Delta pc_{i,j} \tag{4}$$

## 3.3 Neural Network Integration

The ESM module can be integrated anywhere into a wider CNN stack, forming an Egospheric Spatial Memory Network (ESMN). Throughout this paper we consider two variants, ESMN and ESMN-RGB, see Figure 2. ESMN-RGB is a special case of ESMN, where RGB features are directly projected into memory, while ESMN projects CNN encoded features into memory. The inclusion of polar angles, azimuthal angles and depth means the full relative polar coordinates are explicitly represented for each pixel in memory. Although the formulation described in Algorithm 1 and Fig 1 allows for $m$ vision sensors, the experiments in this paper all involve only a single acquiring sensor, meaning $m = 1$. We also only consider cases with constant variance in the acquired images $V_t = k_{var}$, and so we omit the variance images from the ESM input in Fig 2 for simplicity. For baseline approaches, we compute an image of camera-relative coordinates via $K^{-1}$, and then concatenate this to the RGB image along with the tiled incremental poses before input to the networks. All values are normalized to $0 - 1$ before passing to convolutions, based on the permitted range for each channel.

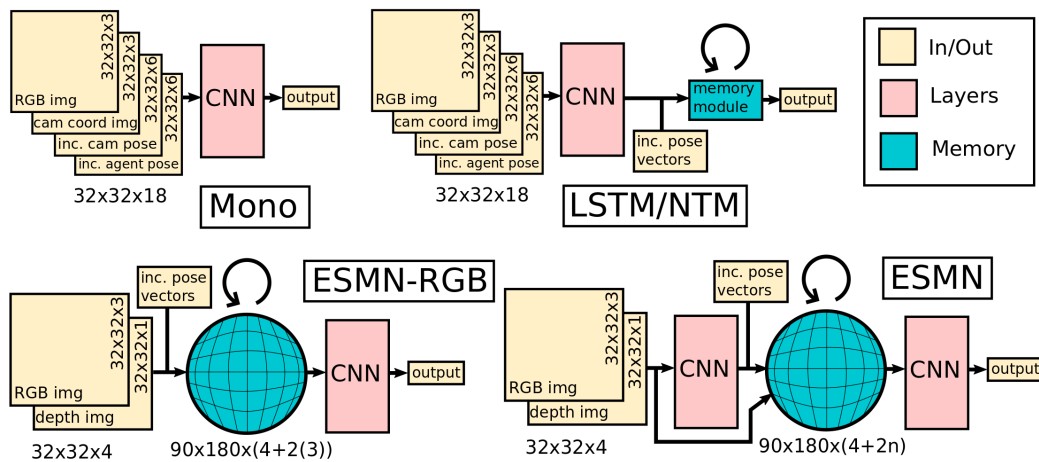

Figure 2: High level schematics of the ESM-integrated network architectures ESMN-RGB and ESMN, as well as other baseline architectures used in the experiments: Mono, LSTM and NTM.

## 4 EXPERIMENTS

The goal of our experiments is to show the wide applicability of ESM to different embodied 3D learning tasks. We test two different applications:

1. Image-to-action learning for multi-DOF control (Sec 4.1). Here we consider drone and robot manipulator target reacher tasks using either ego-centric or scene-centric cameras. We then assess the ability for ESMN policies to generalize between these different camera modalities, and assess the utility of the ESM geometry for obstacle avoidance. We train policies both using imitation learning (IL) and reinforcement learning (RL).

2. Object segmentation (Sec 4.2). Here we explore the task of constructing a semantic map, and the effect of changing the ESM module location in the computation graph on performance.

### 4.1 MULTI-DOF VISUOMOTOR CONTROL

While ego-centric cameras are typically used when learning to navigate planar scenes from images (Jaderberg et al., 2016; Zhu et al., 2017; Gupta et al., 2017; Parisotto & Salakhutdinov, 2017), *static* scene-centric cameras are the de facto when learning multi-DOF controllers for robot manipulators (Levine et al., 2016; James et al., 2017; Matas et al., 2018; James et al., 2019b). We consider the more challenging and less explored setup of learning multi-DOF visuomotor controllers from ego-centric cameras, and also from *moving* scene-centric cameras. LSTMs are the de facto memory architecture in the RL literature (Jaderberg et al., 2016; Espeholt et al., 2018; Kapturowski et al., 2018; Mirowski et al., 2018; Bruce et al., 2018), making this a suitable baseline. NTMs represent another suitable baseline, which have outperformed LSTMs on visual navigation tasks (Wayne et al., 2018). Many other works exist which outperform LSTMs for planar navigation in 2D maze-like environments (Gupta et al., 2017; Parisotto & Salakhutdinov, 2017; Henriques & Vedaldi, 2018), but the top-down representation means these methods are not readily applicable to our multi-DOF control tasks. LSTM and NTM are therefore selected as competitive baselines for comparison.

### 4.1.1 IMITATION LEARNING

For our imitation learning experiments, we test the utility of the ESM module on two simulated visual reacher tasks, which we refer to as Drone Reacher (*DR*) and Manipulator Reacher (*MR*). Both are implemented using the CoppeliaSim robot simulator (Rohmer et al., 2013), and its Python extension PyRep (James et al., 2019a). We implement DR ourselves, while MR is a modification of the reacher task in RLBench (James et al., 2020). Both tasks consist of 3 targets placed randomly in a simulated arena, and colors are newly randomized for each episode. The targets consist of a cylinder, sphere, and "star", see Figure 3.

In both tasks, the target locations remain fixed for the duration of an episode, and the agent must continually navigate to newly specified targets, reaching as many as possible in a fixed time frame of 100 steps. The targets are specified to the agent either as RGB color values or shape class id, depending on the experiment. The agent does not know in advance which target will next be specified, meaning a memory of all targets and their location in the scene must be maintained for the full duration of an episode. Both environments have a single body-fixed camera, as shown in Figure 3, and also an external camera with freeform motion, which we use separately for different experiments.

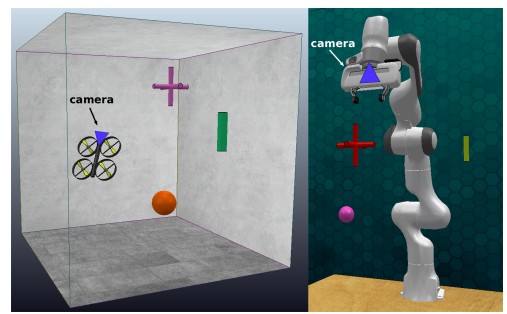

Figure 3: Visualization of (a) Drone Reacher and (b) Manipulator Reacher tasks.

For training, we generate an offline dataset of 100k 16-step sequences from random motions for both environments, and train the agents using imitation learning from known expert actions. Action spaces of joint velocities $\dot{q} \in \mathbb{R}^7$ and cartesian velocities $\dot{x} \in \mathbb{R}^6$ are used for *MR* and *DR* respectively. Expert translations move the end-effector or drone directly towards the target, and expert rotations rotate the egocentric camera towards the target via shortest rotation. Expert joint velocities are calculated for linear end-effector motion via the manipulator Jacobian. For all experiments, we compare to baselines of single-frame, dual-stacked LSTM with and without spatial auxiliary losses, and NTM. We also compare against a network trained on partial oracle omni-directional images, masked at unobserved pixels, which we refer to as Partial-Oracle-Omni (PO2), as well as random and expert policies. PO2 cannot see regions where the monocular camera has not looked, but it maintains a pixel-perfect memory of anywhere it has looked. Full details of the training setups are provided in Appendix A.1. The results for all experiments are presented in Table 1.

| | Drone Reacher | | | | Manipulator Reacher | | | |
|---|---|---|---|---|---|---|---|---|
| | Ego Acq | | Freeform Acq | | Ego Acq | | Freeform Acq | |
| | Color | Shape | Color | Shape | Color | Shape | Color | Shape |
| Mono | 0.6(0.7) | 0.9(1.7) | 2.4(5.0) | 0.5(1.6) | 1.8(1.5) | 1.6(1.1) | 0.1(0.2) | 0.1(0.2) |
| LSTM | 12.7(3.4) | 4.1(2.3) | 1.0(1.0) | 0.6(0.8) | 1.0(0.5) | 0.1(0.2) | 0.1(0.2) | 0.1(0.4) |
| LSTM Aux | 1.3(0.8) | 0.4(0.8) | 2.4(2.2) | 1.9(1.7) | 1.0(0.7) | 0.1(0.3) | 0.3(0.6) | 0.0(0.2) |
| NTM | 10.5(4.2) | 2.5(1.9) | 3.2(2.9) | 1.6(1.5) | 1.0(0.6) | 0.2(0.4) | 0.1(0.3) | 0.1(0.2) |
| ESMN-RGB | **20.6**(7.3) | 4.1(3.8) | **16.1**(12.7) | 1.1(2.6) | **11.4**(5.1) | 3.1(3.2) | **10.5**(5.7) | **0.9**(1.6) |
| ESMN | **20.8**(7.8) | **18.3**(6.4) | **16.6**(12.9) | **8.5**(11.2) | **11.7**(5.3) | **4.7**(4.0) | **11.0**(5.8) | **1.0**(1.2) |
| Random | 0.1(0.2) | 0.1(0.2) | 0.1(0.2) | 0.1(0.2) | 0.1(0.2) | 0.1(0.2) | 0.1(0.2) | 0.1(0.2) |
| PO2 | 21.0(8.6) | 14.4(6.1) | 19.1(12.7) | 3.9(8.1) | 13.0(5.9) | 4.1(3.5) | 12.5(6.1) | 2.6(2.5) |
| Expert | 21.3(8.4) | 21.3(8.4) | 21.3(8.4) | 21.3(8.4) | 16.5(5.2) | 16.5(5.2) | 16.5(5.2) | 16.5(5.2) |

Table 1: Final policy performances on the various drone reacher (DR) and manipulator reacher (MR) tasks, from egocentric acquired (Ego Acq) or freeform acquired (Freeform Acq) cameras, with the network conditioned on either target color or shape. The values indicate the mean number of targets reached in the 100 time-step episode, and the standard deviation, when averaged over 256 runs. ESMN-RGB stores RGB features in memory, while ESMN stores learnt features.

**Ego-Centric Observations:** In this configuration we take observations from body-mounted cameras. We can see in Table 1 that for both DR and MR, our module significantly outperforms other memory baselines, which do not explicitly incorporate geometric inductive bias. Clearly, the baselines have difficulty in optimally interpreting the stream of incremental pose measurements and depth. In contrast, ESM by design stores the encoded features in memory with meaningful indexing. The ESM structure ensures that the encoded features for each pixel are aligned with the associated relative polar translation, represented as an additional feature in memory. When fed to the post-ESM convolutions, action selection can then in principle be simplified to target feature matching, reading the associated relative translations, and then transforming to the required action space. A collection of short sequences of the features in memory for the various tasks are presented in Figure 4, with (a), (b) and (d) coming from egocentric observations. In all three cases we see the agent reach one target by the third frame, before re-orienting to reach the next.

We also observe that ESMN-RGB performs well when the network is conditioned on target color, but fails when conditioned on target shape id. This is to be expected, as the ability to discern shape from the memory is strongly influenced by the ESM resolution, quantization holes, and angular distortion. For example, the "star" shape in Figure 4 (a) is not apparent until $t_5$. However, ESMN is able to succeed, and starts motion towards this star at $t_3$. The pre-ESM convolutional encoder enables ESMN to store useful encoded features in the ESM module from monocular images, within which the shape was discernible. Figure 4 (a) shows the 3 most dominant ESM feature channels projected to RGB.

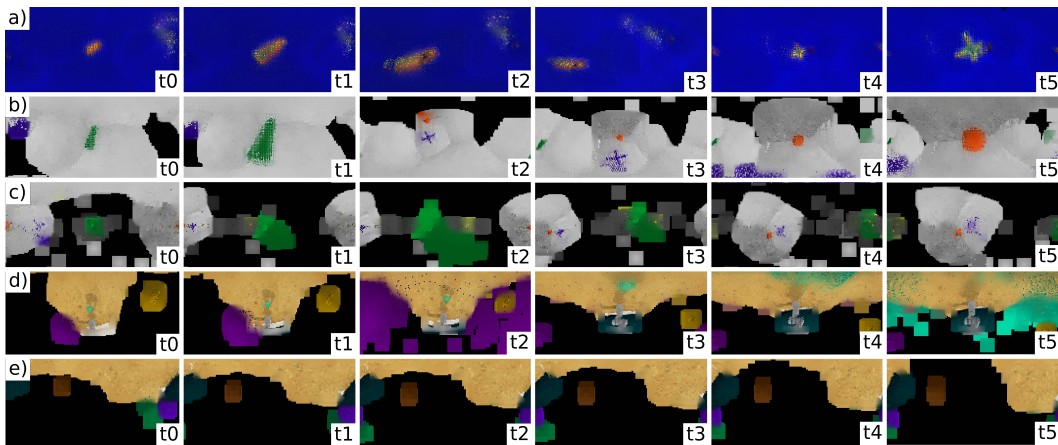

Figure 4: Sample trajectories through the memory for (a) ESMN on DR-Ego-Shape, and ESMN-RGB on (b) DR-Ego-Color, (c) DR-Freeform-Color, (d) MR-Ego-Color, (e) MR-Freeform-Color. The images each correspond to features in the full $90 \times 180$ memory at that particular timestep $t$.

**Scene-Centric Observations:** Here we explore the ability of ESM to generalize to unseen camera poses and motion, from cameras external to the agent. The poses of these cameras are randomized for each episode during training, and follow random freeform rotations, with a bias to face towards the centre of the scene, and linear translations. Again, we see that the baselines fail to learn successful policies, while ESM-augmented networks are able to solve the task, see Table 1. The memories in these tasks take on a different profile, as can be seen in Fig 4 (c) and (e). While the memories from egocentric observations always contain information in the memory image centre, where the most recent monocular frame projects with high density, this is not the case for projections from arbitrarily positioned cameras which can move far from the agent, resulting in sparse projections into memory. The targets in Fig 4 (e) are all represented by only 1 or 2 pixels. The large apparent area in memory is a result of the variance-based smoothing, where the low-variance colored target pixels are surrounded by high-variance unobserved pixels in the ego-sphere.

**Obstacle Avoidance:** To further demonstrate the benefits of a local spatial geometric memory, we augment the standard drone reacher task with obstacles, see Figure 5. Rather than learning the avoidance in the policy, we exploit the interpretable geometric structure in ESM, and instead augment the policy output with a local avoidance component. We then compare targets reached and collisions for different avoidance baselines, and test these avoidance strategies on random, ESMN-RGB and expert target reacher policies. We see that the ESM geometry enables superior avoidance over using the most recent depth frame alone. The obstacle avoidance results are presented in Table 2, and further details of the experiment are presented in Appendix A.2.

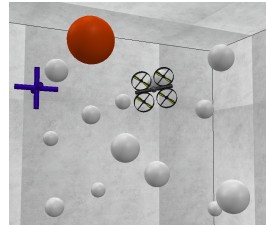

Figure 5: Avoidance task

**Camera Generalization:** We now explore the extent to which policies trained from egocentric observations can generalize to cameras moving freely in the scene, and vice-versa. The results of these transfer learning experiments are presented in Table 3. Rows not labelled "Transferred" are taken directly from Table 1, and repeated for clarity. Example image trajectories for both egocentric and free-form observations are presented in Figure 6. The trained networks were not modified in any way, with no further training or fine-tuning applied before evaluation on the new image modality.

| | | Policy | | | | | |
|---|---|---|---|---|---|---|---|
| | | Random | | ESMN-RGB | | Expert | |
| | | Reached | Collisions | Reached | Collisions | Reached | Collisions |
| Avoidance | No Avoidance | 0.0(0.2) | 29.4(27.9) | 9.9(7.0) | 64.8(67.4) | 21.3(2.1) | 121.7(15.8) |
| | Single Depth Frame | 0.1(0.2) | 9.6(11.8) | 11.1(4.3) | 10.8(13.0) | 15.3(2.0) | 7.7(6.9) |
| | ESM Depth Map | 0.1(0.3) | **4.2**(7.1) | 9.6(4.5) | **2.7**(5.5) | 10.8(4.4) | **2.2**(3.8) |
| | Ground Truth | 0.1(0.3) | 0.1(0.5) | 9.6(5.1) | 0.0(0.0) | 15.2(3.9) | 0.0(0.0) |

Table 2: Targets reached and number of collision for the obstacle avoidance drone task. Full details of this experiment are provided in Appendix A.2.

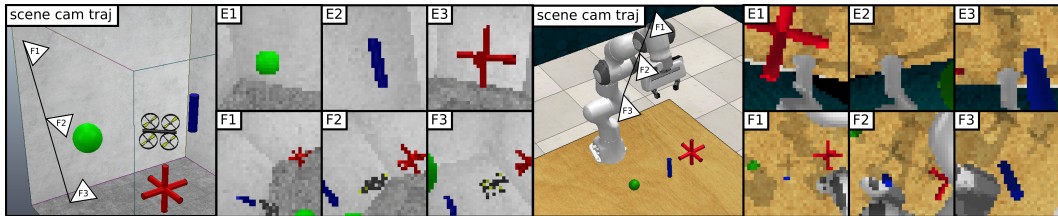

Figure 6: Example image sequences from both egocentric (E) and freeform (F) cameras, on both reacher tasks. The images are all time-aligned, and correspond to the same agent motion in the scene.

| | Drone Reacher | | | | Manipulator Reacher | | | |
|---|---|---|---|---|---|---|---|---|
| | Ego Acq | | Freeform Acq | | Ego Acq | | Freeform Acq | |
| | Color | Shape | Color | Shape | Color | Shape | Color | Shape |
| LSTM | 12.7(3.4) | 4.1(2.3) | 1.0(1.0) | 0.6(0.8) | 1.0(0.5) | 0.1(0.2) | 0.1(0.2) | 0.1(0.4) |
| Transferred | 0.3(0.6) | 0.5(0.7) | 0.1(0.3) | 0.3(0.7) | 0.1(0.3) | 0.0(0.2) | 0.0(0.0) | 0.0(0.2) |
| ESMN-RGB | 20.6(7.3) | 4.1(3.8) | 16.1(12.7) | 1.1(2.6) | 11.4(5.1) | 3.1(3.2) | 10.5(5.7) | 0.9(1.6) |
| Transferred | **20.3**(11.5) | **5.8**(4.7) | **15.3**(10.9) | 0.8(1.6) | **11.2**(5.6) | **3.9**(4.3) | **9.8**(3.8) | 1.1(1.3) |
| ESMN | 20.8(7.8) | 18.3(6.4) | 16.6(12.9) | 8.5(11.2) | 11.7(5.3) | 4.7(4.0) | 11.0(5.8) | 1.0(1.2) |
| Transferred | **10.4**(5.5) | 0.7(0.9) | **8.9**(6.5) | 1.0(1.4) | **7.5**(4.9) | **6.3**(6.2) | **6.8**(4.7) | 0.6(1.0) |

Table 3: Reacher performances both with and without camera transfer, using the same notation and setup as described in Table 1. Successful transfers are highlighted in bold.

### 4.1.2 REINFORCEMENT LEARNING

Assuming expert actions in partially observable (PO) environments is inherently limited. It is not necessarily true that the best action always rotates the camera directly to the next target for example. In general, for finding optimal policies in PO environments, methods such as reinforcement learning (RL) must be used. We therefore train both ESM networks and all the baselines on a simpler variant of the MR-Ego-Color task via DQN (Mnih et al., 2015). The manipulator must reach red, blue and then yellow spherical targets from egocentric observations, after which the episode terminates. We refer to this variant as MR-Seq-Ego-Color, due to the sequential nature. The only other difference to MR is that MR-Seq uses $128 \times 128$ images as opposed to $32 \times 32$. The ESM-integrated networks again outperform all baselines, learning to reach all three targets, while the baseline policies all only succeed in reaching one. Full details of the RL setup and learning curves are given in Appendix A.3.

### 4.2 OBJECT SEGMENTATION

We now explore the suitability of the ESM module for object segmentation. One approach is to perform image-level segmentation in individual monocular frames, and then perform probabilistic fusion when projecting into the map (McCormac et al., 2017). We refer to this approach as Mono. Another approach is to first construct an RGB map, and then pass this map as input to a network. This has the benefit of wider context, but lower resolution is necessary to store a large map in memory, meaning details can be lost. ESMN-RGB adopts this approach. Another approach is to combine monocular predictions with multi-view optimization to gain the benefits of wider surrounding context as in (Zhi et al., 2019). Similarly, the ESMN architecture is able to combine monocular inference with

the wider map context, but does so by constructing a network with both image-level and map-level convolutions. These ESM variants adopt the same broad architectures as shown in Fig 2, with the full networks specified in Appendix A.4. We do not attempt to quote state-of-the-art results, but rather to further demonstrate the wide applications of the ESM module, and to explore the effect of placing the ESM module at different locations in the convolutional stack. We evaluate segmentation accuracy based on the predictions projected to the ego-centric map. With fixed network capacity between methods, we see that ESMN outperforms both baselines, see Table 4 for the results, and Figure 7 for example predictions in a ScanNet reconstruction. Further details are given in Appendix A.4.

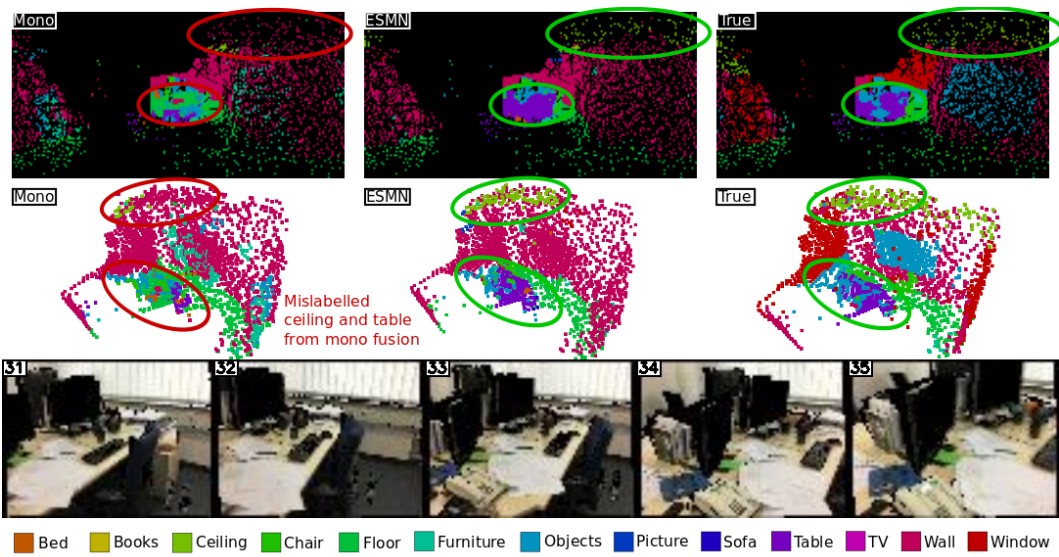

Figure 7: Top: Segmentation predictions in ESM memory for Mono and ESMN, and ground truth. Middle: Point cloud rendering of the predictions. Bottom: Monocular input image sequence.

| | Images $60 \times 80$ | | Images $120 \times 160$ | | Images $60 \times 80$ | |
| | Memory $90 \times 180$ | | Memory $90 \times 180$ | | Memory $180 \times 360$ | |
| | 1-frame | 16-frame | 1-frame | 16-frame | 1-frame | 16-frame |
|---|---|---|---|---|---|---|
| Mono | **54.5**(19.0) | 55.1(13.6) | **54.9**(19.3) | 55.6(13.9) | **54.6**(19.2) | 55.3(13.7) |
| ESMN-RGB | 51.4(19.6) | 54.1(13.8) | 51.7(19.3) | 54.4(13.2) | **54.0**(19.2) | 57.1(13.5) |
| ESMN | **55.0**(19.0) | **59.4**(12.7) | **55.3**(19.1) | **59.8**(13.0) | **55.2**(19.4) | **59.7**(12.9) |

Table 4: Object segmentation segmentation accuracies on the ScanNet test set for a monocular fusion baseline (Mono), as well as ESMN-RGB and ESMN.

## 5 CONCLUSION

Through a diverse set of demonstrations, we have shown that ESM represents a widely applicable computation graph and trainable module for tasks requiring general spatial reasoning. When compared to other memory baselines for image-to-action learning, our module outperforms these dramatically when learning both from ego-centric and scene-centric images. One weakness of our method is that is assumes the availability of both incremental pose measurements of all scene cameras, and depth measurements, which is not a constraint of the other memory baselines. However, with the ever increasing ubiquity of commercial depth sensors, and with there being plentiful streams for incremental pose measurements including visual odometry, robot kinematics, and inertial sensors, we argue that such measurements are likely to be available in any real-world application of embodied AI. We leave it to future work to investigate the extent to which ESM performance deteriorates with highly uncertain real-world measurements, but with full uncertainty propagation and probabilistic per pixel fusion, ESM is well suited for accumulating noisy measurements in a principled manner.

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

# A    APPENDIX

## A.1    MULTI-DOF IMITATION LEARNING

### A.1.1    OFFLINE DATASETS

The image sequences for the offline datasets are captured following random motion of the agent in both the DR and MR tasks, but known expert actions for each of the three possible targets in the scene are stored at every timestep. The drone reacher is initialized in random locations at the start of each episode, whereas the manipulator reacher is always started in the same robot configuration overlooking the workspace, as in the original RLBench reacher task.

For the scene-centric acquisition, we instantiate three separate scene-centric cameras in the scene. In order to maximise variation in the dataset to encourage network generalization to arbitrary motions at test-time, we reset the pose of each of these scene-centric cameras at every step of the episode, rather than having each camera follow smooth motions. Each new random pose has a rotational bias to face towards the scene-centre, to ensure objects are likely to be seen frequently. By resetting the cameras poses on every time-step, we encourage the networks to learn to make sense of the pose information given to the network, rather than learning policies which fully rely on smoothly and slowly varying images.

Both tasks use ego-centric cameras with a wider field of view (FOV) than the scene-centric cameras. This is a common choice in robotics, where wide angle perception is especially necessary for body-mounted cameras. RLBench by default uses ego-centric FOV of 60 degrees and scene-centric FOV of 40 degrees, and we use the same values for our RLBench-derived MR task. For the drone reacher task, we use ego-centric FOV of 90 degrees and scene-centric FOV of 55 degrees, to enable all methods to more quickly explore the perceptual ego-sphere.

For the manipulator reacher dataset, we also store a robot mask image, which shows the pixels corresponding to the robot for all ego-centric and scene-centric images acquired. Known robot forward kinematics are used for generating the masking image. All images in the offline dataset are $32 \times 32$ resolution.

### A.1.2 TRAINING

To maximize the diversity from the offline datasets, a new target is randomly chosen from each of the three possible targets for each successive step in the unrolled time dimension in the training batch. This ensures maximum variation in sequences at train-time, despite a relatively small number of 100k 16-frame sequences stored in the offline dataset. This also strongly encourages each of the networks to learn to remember the location of every target seen so far, because any target can effectively be requested from the training loss at any time.

Similarly, for the scene-centric cameras we randomly choose one of the three scene cameras at each successive time-step in the unrolled time dimension for maximum variation. Again this forces the networks to make use of the camera pose information to make sense of the scene, and prevents overfitting on particular repeated sequences in the training set, instead encouraging generalization to fully arbitrary motions. For these experiments, the baseline methods of Mono, LSTM, LSTM-Aux and NTM also receive the full absolute camera pose at each step rather than just incremental poses received by ESM, as we found this to improve the performance of the baselines.

For training the manipulator reacher policies, we additionally use the robot mask images to set high variance pixel values before feeding to the ESM module. This prevents the motion of the robot from breaking the static-scene assumption adopted by ESM during re-projections. We also provide the mask image as input to the baselines.

All networks use a batch size of 16, an unroll size of 16, and are trained for 250k steps using an ADAM optimizer with $1e-4$ learning rate. None of the memory baselines cache the the internal state between training batches, and so the networks must learn to utilize the memory during the 16-frame unroll. 16 frames is on average enough steps to reach all 3 of the targets for both tasks.

### A.1.3 NETWORK ARCHITECTURES

The network architectures used in the imitation learning experiments are provided in Fig 8. Both LSTM baselines use dual stacked architectures with hidden and cell state sizes of $1024$. For NTM we use a similar variant to that used by Wayne et al. (2018), namely, we use sequential writing to the memory, and retroactive updates. Regarding the 16-frame unroll, we again emphasize that 16 steps is on average enough time to reach all targets once. In order to encourage generalization to longer sequences than only 16-steps, we limit the writable memory size to 10, and track the usage of these 10 memory cells with a *usage indicator* such that subsequent writes can preferentially overwrite the least used of the 10 cells. This again is the same approach used by Wayne et al. (2018), which is one of very few works to successfully apply NTM-style architectures to image-to-action domains. It's important to note that the use of retroactive updates makes the *total* memory size actually 20, as half of the cells are always reserved for the retroactive memory updates. Regarding image padding at the borders for input to the convolutions, the Mono and LSTM/NTM baselines use standard zero padding, whereas ESMN-RGB and ESMN pad the outer borders with the wrapped omni-directional image.

### A.1.4 AUXILIARY LOSSES

Motivated by the fact that many successful applications of LSTMs to image-to-actions learning involve the use of spatial auxiliary losses (Jaderberg et al., 2016; James et al., 2017; Sadeghi et al., 2017; Mirowski et al., 2018), we also compare to an LSTM which uses two such auxiliary proposals, namely the attention loss proposed in (Sadeghi et al., 2017) and a 3-dimensional Euler-based variant

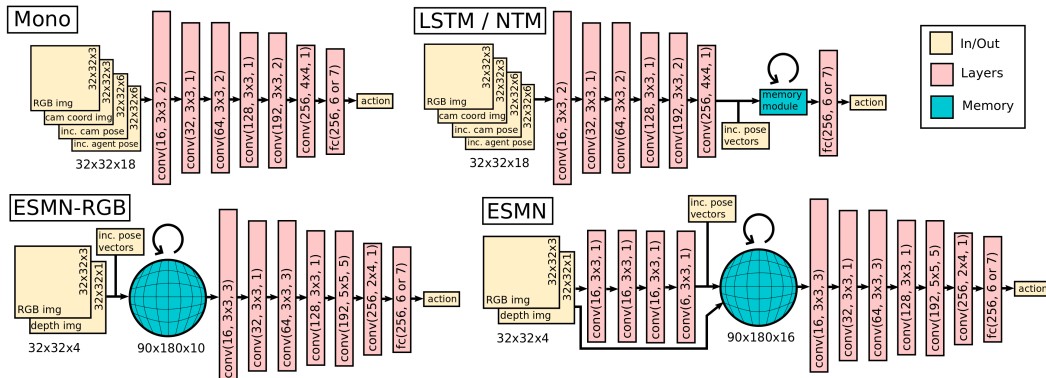

Figure 8: Image-to-action imitation learning network architectures for Mono, LSTM/NTM, ESMN-RGB and ESMN.

of the heading loss proposed in (Mirowski et al., 2018), which itself only considers 1D rotations normal to the plane of navigation. Our heading loss does not compute the 1D rotational offset from North, as this is not detectable from the image input. Instead, the networks are trained to predict the 3D Euler offset from the orientation of the first frame in the sequence. The modified LSTM network architecture is presented in Fig 9, and example images and classification targets for the auxiliary attention loss are presented in Fig 10. We emphasize that we did not attempt to tune these auxiliary losses, and applied them unmodified to the total loss function, taking the mean of the cross entropy loss for each, and linearly scaling so that the total auxiliary loss is roughly the same magnitude as the imitation loss at the start of training. Tuning auxiliary losses on different tasks is known to be challenging, and the losses can worsen performance without time-consuming manual tuning, as evidenced in the performance of the UNREAL agent (Jaderberg et al., 2016) compared to a vanilla RL-LSTM network demonstrated in (Wayne et al., 2018). We reproduce this general finding, and see that the untuned auxiliary losses do not improve performance on our reacher tasks. To further investigate the failure mechanism, we plot the two auxiliary losses on the validation set during training for each task in Fig 11. We find that the heading loss over-fits on the training set in all tasks, without learning any useful notion of incremental agent orientation. This is particularly evidenced in the DR tasks, which start each sequence with random agent orientations. In contrast, predicting orientation relative to the first frame on the MR task is much simpler because the starting pose is always constant in the scene, and so cues for the relative orientation are available from individual frames. This is why we observe a lower heading loss for the MR task variants in Fig 11. We do however still observe overfitting in the MR task. This overfitting on all tasks helps to explain why LSTM-Aux performs worse than the vanilla LSTM baseline for some of the tasks in Table 1. In contrast, the ESM module embeds strong spatial inductive bias into the computation graph itself, requires no tuning at all, and consistently leads to successful policies on the different tasks, with no sign of overfitting on any of the datasets, as we further discuss in Section A.1.5.

### A.1.5 FURTHER DISCUSSION OF RESULTS

The losses for each network evaluated on the training set and validation set during the course of training are presented in Fig 12. We first consider the results for the drone reacher task. Firstly, we can clearly see from the DR-ego-rgb and DR-freeform-rgb tasks that the baselines struggle to interpret the stream of incremental pose measurements and depth, in order to select optimal actions in the training set, and this is replicated in the validation set, and in the final task performance in Tab 1. We can also see that ESMN is able to achieve lower training and validation losses than ESMN-RGB when conditioned on shape in the DR-ego-shape and DR-freeform-shape tasks, and also expectedly achieves higher policy performance, shown in in Tab 1. What we also observe is that the baselines have a higher propensity to over-fit on training data. Both the LSTM and NTM baselines achieve lower training set error than ESMN on the DR-ego-shape task, but not lower validation error. In contrast, all curves for ESM-integrated networks are very similar between the training and validation set. The ESM module by design performs principled spatial computation, and so these networks are inherently much more robust to overfitting.

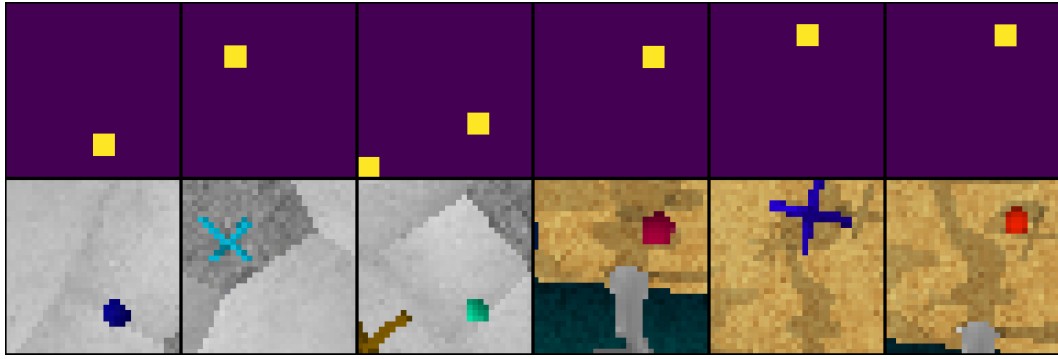

Figure 10: Top: pixel-wise attention classification targets for the LSTM-Aux network on DR (left) and MR (right) tasks. Bottom: the corresponding monocular images from the DR (left) and MR (right) tasks.

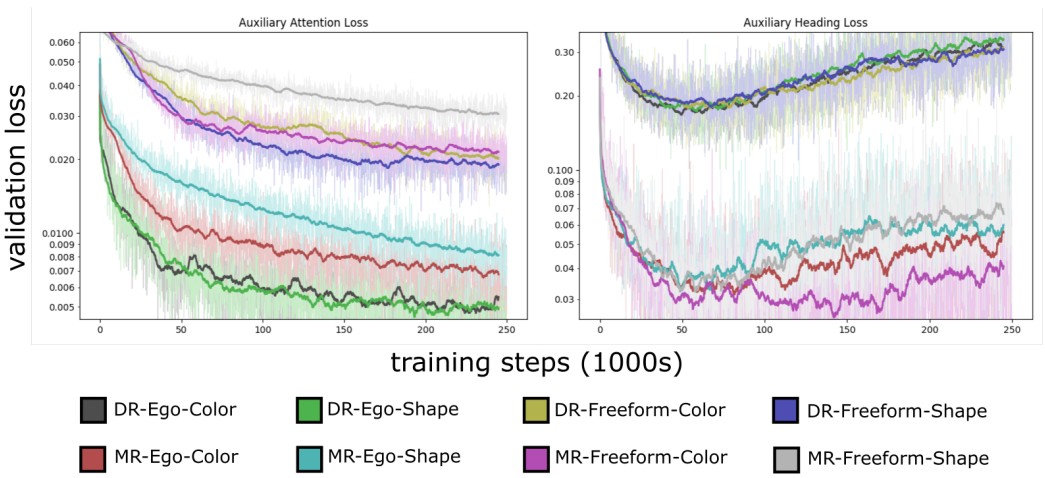

Figure 11: Auxiliary attention (left) and heading (right) losses on the validation set during training for each of the different imitation learning reacher tasks.

Looking to the manipulator reacher task, we first notice that the LSTM and NTM baselines are actually able to achieve lower losses than the ESM-integrated networks on both the training set and validation set for the MR-ego-rgb and MR-ego-shape tasks. However, this does not translate to higher policy performance in Table 1. The reason for this is that the RLBench

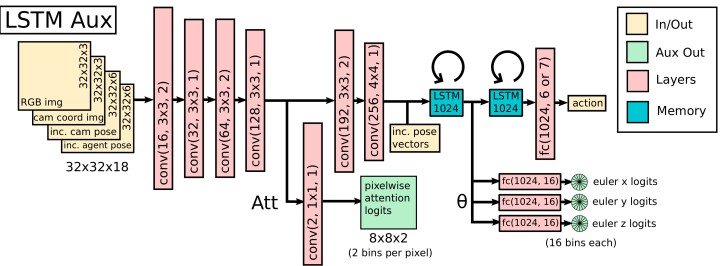

Figure 9: Network architecture for the LSTM-Aux baseline.

reacher task always initializes the robot in the same configuration, and so the diversity in the offline dataset is less than that of the drone reacher offline dataset. The scope of possible robot configurations in each 16-step window in the dataset is more limited. In essence, the baselines are achieving well in both training and validation sets as a result of overfitting to the limited data distributions observed. What these curves again highlight is the strong generalization power of ESM-integrated networks. Despite seeing relatively limited robot configurations in the dataset, the ESM policies

do not overfit on these, and are still able to use this data to learn general policies which succeed from unseen out-of-distribution images at test-time. We also again observe the same superiority of ESMN over ESMN-RGB when conditioned on shape input in the training and validation losses for the MR-ego-shape task.

A final observation is that all methods fail to perform well on the MR-freeform-shape task. We investigated this, and the weak performance is a combined result of low-resolution $32 \times 32$ images acquired in the training set and the large distance between the scene-centric cameras and the targets in the scene. The shapes are often difficult to discern from the monocular images acquired, and so little information is available for the methods to successfully learn a policy. We expect that with higher resolution images, or with an average lower distance between the scene-cameras and workspace in the offline dataset, we would again observe the ESM superiority observed for all other tasks.

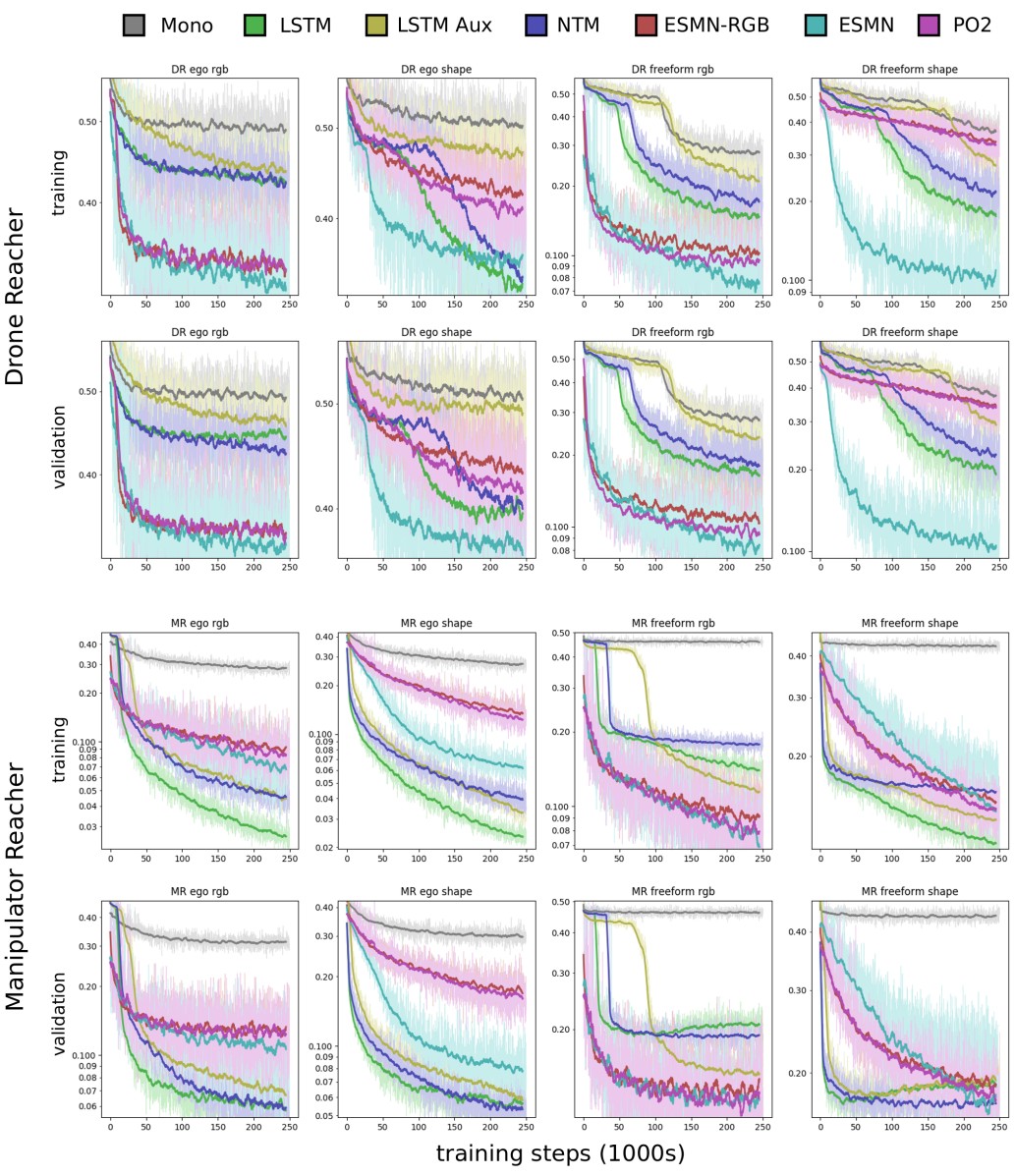

Figure 12: Network losses on the training set (top 8) and validation set (bottom 8) during the course of training for imitation learning from the offline datasets, for the different reacher task.

### A.1.6 Implicit Feature Analysis

Here we briefly explore the nature of the the features which the end-to-end ESM module learns to store in memory for the different reacher tasks. For each task, we perform a Principal Component Analysis (PCA) on the features encoded by the pre-ESM encoders in the ESMN networks. We compute the principal components (PCs) using encoded features from all monocular images in the training dataset. We present example activations for each of the 6 principal components for a sample of monocular images taken from each of the shape conditioned task variations in in Fig 13, with the most dominate principal components shown on the left in green, going through to the least dominant principal component on the right in purple. Each principal component is projected to a different color-space for better visualization, with plus or minus one standard deviation of the principal component mapping to the full color-space. Lighter colors correspond to higher PC activations.

We can see that most dominant PC (shown in green) for the drone reacher tasks predominantly activate for the background, and the third PC (blue) appears to activate most strongly for edges. The principal components also behave similarly on the MR-Ego-Shape task. However, on the MR-Freeform-Shape task, which neither ESMN nor any of the baselines are able to succeed on, the first PC appears to activate strongly on both the arm and the target shapes.

The main conclusion which we can draw from Fig 13 is that the pre-ESM encoder does not directly encode shape classes as might be expected. Instead, the encoder learns to store other lower level features into ESM. However, as evidenced in the results in Table 1, the combination of these lower level features in ESM is clearly sufficient for the post-ESM convolutions to infer the shape id for selecting the correct actions in the policy, at least by using a collection of the encoded features within a small receptive field, which is not possible when using pure RGB features.

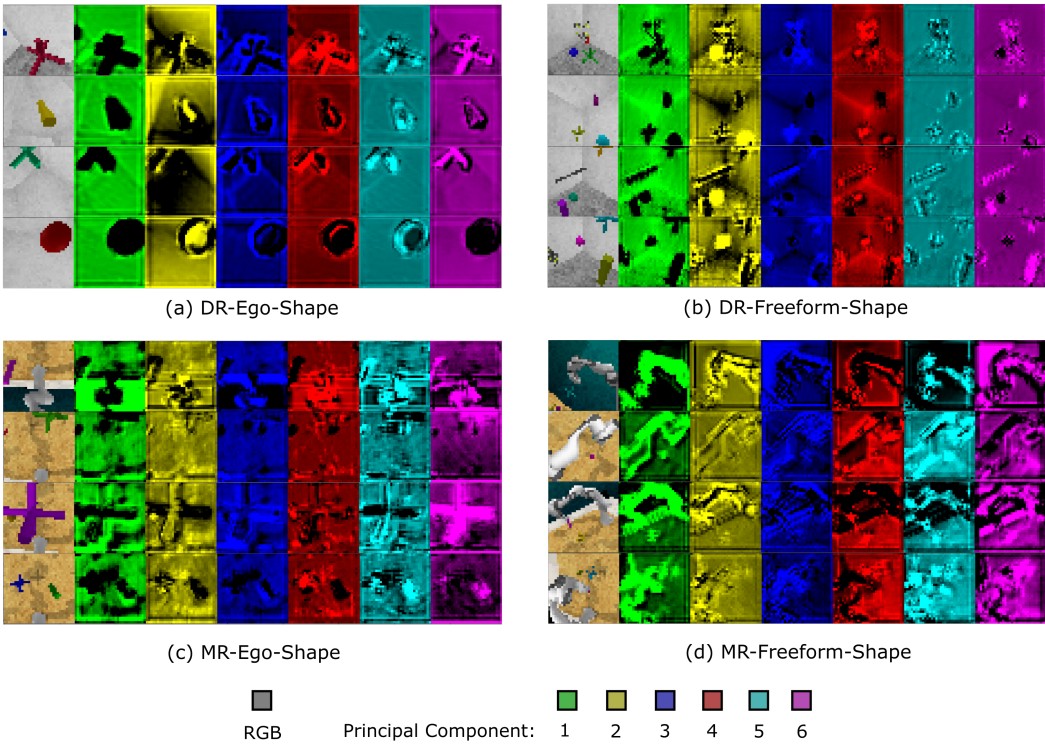

Figure 13: Principal Components (PCs) of the features from the pre-ESM encoder of the ESMN architecture on some example images for each of the four shape-conditioned reacher tasks, with each of the six the principal components mapped to different colors to maximise clarity. PCs go from most dominant on the left (green) to least dominant on the right (purple). Lighter values correspond to higher PC activation, with black indicating low activation.

## A.2 OBSTACLE AVOIDANCE

For this experiment, we increase the drone reacher environment by $2\times$ in all directions, resulting in an $8\times$ increase in volume. We then also add 20 spherical obstacles into the scene with radius $r = 0.1m$. For the avoidance, we consider a bubble around the agent with radius $R = 0.2$, and flag a collision whenever any part of an obstacle enters this bubble. Given the closest depth measurement available $d_{closest}$, the avoidance algorithm simply computes an avoidant velocity vector $v_a$ whose magnitude is inversely proportional to the distance from collision, clipped to the maximum velocity $|v|_{max}$. Equation 5 shows the calculation for the avoidance vector magnitude. We run the avoidance controller at $10\times$ the rate of the ESM updates.

$$|v_a| = min\left[\frac{10^{-3}}{max\left[d_{closest} - R, 10^{-12}\right]^2}, |v|_{max}\right] \qquad (5)$$

In order to prevent avoidant motion away from the targets to reach, we retrain the ESMN-RGB networks on the drone reacher task, but we train the network to also predict the full relative target location as an additional auxiliary loss. When evaluating on the obstacle avoidance task, we prevent depth values within a fixed distance of this predicted target location from influencing the obstacle avoidance. This has the negative effect of causing extra collisions when the agent erroneously predicts that the target is close, but it enables the agent to approach and reach the target without being pushed away by the avoidance algorithm. Regarding the performance against the baseline, we re-iterate that all monocular images have a large field-of-view of 90 degrees, and yet we still observe significant reductions in collisions when using the full ESM geometry for avoidance, see Tab 2.

## A.3 MULTI-DOF REINFORCEMENT LEARNING

### A.3.1 TRAINING

For the reinforcement learning experiment, we train both ESMN and ESMN-RGB as well as all baselines on a similar sequential target reacher task as defined in Section 4.1 via DQN (Mnih et al., 2015), where the manipulator must reach red, blue and then yellow targets from egocentric observations. We use $(128 \times 128)$ images in this experiment rather than $(32 \times 32)$ as used in the imitation learning experiments. We also use an unroll length of 8 rather than 16. We use discrete delta end-effector translations, with $\pm 0.05$ meters for each axis, with no rotation (resulting in an action size of 6). We use a shaped reward of $r = -len(remaining\_targets) - \|e - g\|_2$, where $e$ and $g$ are the gripper and current target translation respectively.

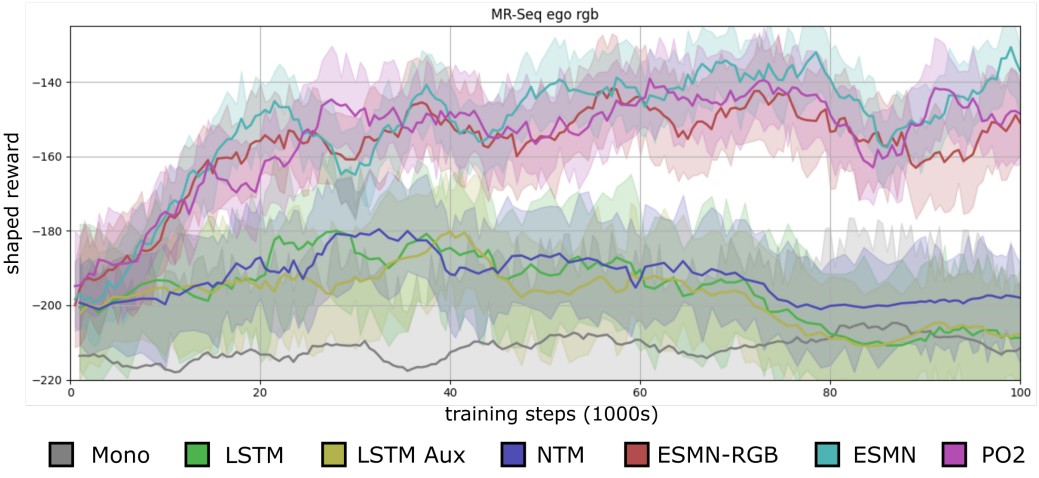

Figure 14: Average return during RL training on sequential reacher task over 5 seeds. Shaded region represent the $min$ and $max$ across trials.

### A.3.2 NETWORK ARCHITECTURES

In order to make the RL more tractable, and enable larger batch sizes, we use smaller network than used in the imitation learning experiments. Both methods use a Siamese network to process the RGB and coordinate-image inputs separately, and consist of 2 convolution (conv) layers with 16 and 32 channels (for each branch). We fuse these branches with a 32 channel conv and 1x1 kernel. The remainder of the architecture then follows the same as in the imitation learning experiments, but we instead use channel sizes of 64 throughout. The network outputs 6 values corresponding to the Q-values for each action. All methods use a learning rate of 0.001, target Q-learning $\tau = 0.001$, batch size of 128, and *leakyReLU* activations. We use an epsilon greedy strategy for exploration that is decayed over 100k training steps from 1 to 0.1. We show the average shaped return during RL training on the sequential reacher task over 5 seeds in Fig 14. Both ESM policies succeed in reaching all 3 targets, whereas all baseline approaches generally only succeed in reaching 1 target. The Partial-Oracle-Omni (PO2) baseline also succeeds in reaching all 3 targets.

## A.4 OBJECT SEGMENTATION

### A.4.1 DATASET

For the object segmentation experiment, we use downsampled $60 \times 80$ and $120 \times 160$ images from the Scan-Net dataset, which we first RGB-Depth align. We use a reduced dataset with frame skip of 30 to maximize diversity whilst minimizing dataset memory. Many sequences contain slow camera motion, resulting in adjacent frames which vary very little. We use the Eigen-13 classification labels as training targets.

### A.4.2 NETWORK ARCHITECTURES

The Mono, ESMN-RGB and ESMN networks all exhibit a U-Net architecture, and output object segmentation predictions in an ego-sphere map. ESMN-RGB and ESMN do so with a U-Net connecting the ESM

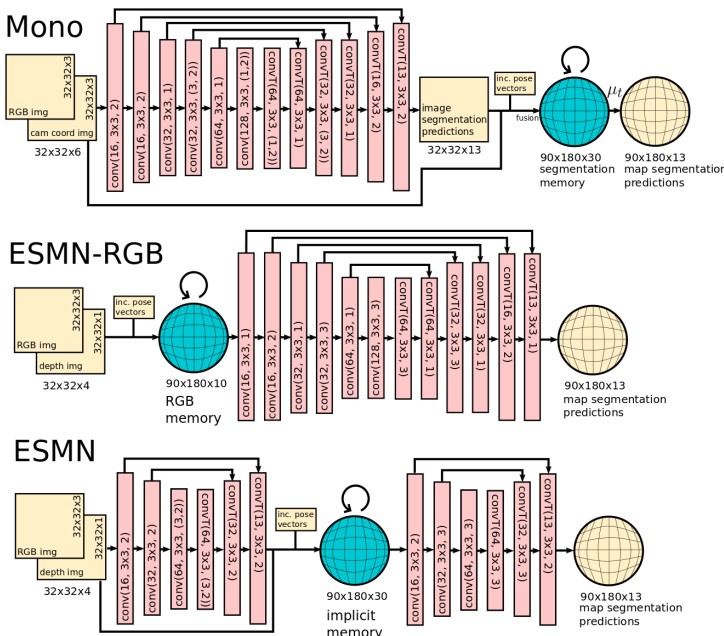

Figure 15: Object segmentation network architectures for Mono, ESMN-RGB and ESMN.

output to the final predictions, and Mono does so by projecting and probabilistically fusing the monocular segmentation predictions in a non-learnt manner. The network architectures are all presented in Fig 15. Regarding image padding at the borders for input to the convolutions, the Mono and LSTM/NTM baselines use standard zero padding, whereas ESMN pads the outer borders with the wrapped omni-directional image.

### A.4.3 TRAINING

For training, all losses are computed in the ego-sphere map frame of reference, either following convolutions for ESMN and ESMN-RGB, or following projection and probabilistic fusion for the Mono case. We compute ground-truth segmentation training target labels by projecting the ground truth monocular frames to form an ego-sphere target segmentation image, see the right-hand-side of

Fig 7 for an example. We chose this approach over computing the ground truth segmentations from the complete ScanNet meshes for implementational simplicity. All experiments use a batch size of 16, unroll size of 8 in the time dimension, and Adam optimizer with learning rate $1e - 4$, trained for 250k steps.

## A.5 RUNTIME ANALYSIS

In this section, we perform a runtime analysis of the ESM memory module. We explore the extent to which inference speed is affected both by monocular resolution and egosphere resolution, as well the differences between CPU and GPU devices, and the choice of machine learning framework. Our ESM module is implemented using Ivy (Lenton et al., 2021), which is a templated deep learning framework supporting multiple backend frameworks. The implementation of our module is therefore jointly compatible with TensorFlow 2.0, PyTorch, MXNet, Jax and Numpy. We analyse the runtimes of both the TensorFlow 2.0 and PyTorch implementations, the results are presented in Tables 5, 6, 7, and 8. All analysis was performed while using ESM with RGB projections to reconstruct ScanNet scene 0002-00 shown in Fig 16. The timing is averaged over the course of the 260 frames in the frame-skipped image sequence, with a frame skip of 30, for this scene. ESM steps with $960 \times 1280$ monocular images were unable to fit into the 11GB of GPU memory when using the PyTorch implementation, and so these results are omitted in Table 8.

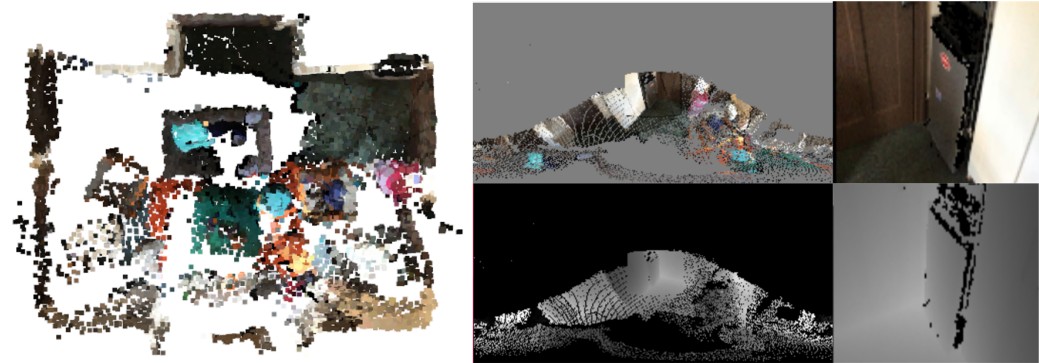

Figure 16: Left: point cloud representation of the ego-centric memory around the camera after full rotation in ScanNet scene 0002-00, with RGB features. Mid: (top) Equivalent omni-directional RGB image, (bottom) equivalent omni-directional depth image, both without smoothing to better demonstrate the quantization holes. Right: (top) A single RGB frame, (bottom) a single depth frame. This is the reconstruction produced during the time-analysis. This particular reconstruction used a monocular resolution of $120 \times 60$, and a memory resolution of $180 \times 360$

|  |  | Monocular Res | | | | |
|---|---|---|---|---|---|---|
|  |  | $60 \times 80$ | $120 \times 160$ | $240 \times 320$ | $480 \times 640$ | $960 \times 1280$ |
| Memory Res | $45 \times 90$ | 245.4 | 162.6 | 83.7 | 24.4 | 6.3 |
|  | $90 \times 180$ | 140.1 | 126.5 | 70.8 | 23.3 | 6.1 |
|  | $180 \times 360$ | 63.9 | 64.0 | 47.5 | 19.2 | 5.8 |
|  | $360 \times 720$ | 16.3 | 14.3 | 14.5 | 11.1 | 4.7 |
|  | $720 \times 1440$ | 3.9 | 3.7 | 3.6 | 3.6 | 2.7 |
|  | $1440 \times 2880$ | 1.1 | 1.1 | 1.0 | 1.0 | 0.9 |

Table 5: Average frames-per-second (fps) runtime for the TensorFlow 2 implemented ESM module on the ScanNet scene 0002-00, with RGB projections, running on 8 CPU cores.

What we see from these runtime results is that the off-the-shelf ESM module is fully compatible as a real-time mapping system. Compared to more computationally intensive mapping and fusion pipelines, the simplicity of ESM makes it particularly suitable for applications where depth and pose measurements are available, and highly responsive computationally cheap local mapping is a strong requirement, such as on-board mapping for drones.

| | | Monocular Res | | | | |
|---|---|---|---|---|---|---|
| | | $60 \times 80$ | $120 \times 160$ | $240 \times 320$ | $480 \times 640$ | $960 \times 1280$ |
| Memory Res | $45 \times 90$ | 262.1 | 223.8 | 166.5 | 76.2 | 24.1 |
| | $90 \times 180$ | 193.5 | 214.8 | 164.4 | 73.5 | 23.6 |
| | $180 \times 360$ | 184.6 | 181.6 | 147.6 | 71.5 | 23.0 |
| | $360 \times 720$ | 91.0 | 89.1 | 83.6 | 56.8 | 21.6 |
| | $720 \times 1440$ | 19.8 | 19.5 | 18.7 | 17.6 | 11.3 |
| | $1440 \times 2880$ | 4.9 | 4.8 | 4.8 | 4.5 | 4.2 |

Table 6: Average frames-per-second (fps) runtime for the TensorFlow 2 implemented and pre-compiled ESM module on the ScanNet scene 0002-00, with RGB projections, running on Nvidia RTX 2080 GPU.

| | | Monocular Res | | | | |
|---|---|---|---|---|---|---|
| | | $60 \times 80$ | $120 \times 160$ | $240 \times 320$ | $480 \times 640$ | $960 \times 1280$ |
| Memory Res | $45 \times 90$ | 98.9 | 45.8 | 26.3 | 6.9 | 1.8 |
| | $90 \times 180$ | 57.0 | 36.8 | 22.3 | 6.6 | 1.7 |
| | $180 \times 360$ | 14.1 | 11.6 | 10.5 | 5.0 | 1.5 |
| | $360 \times 720$ | 5.5 | 5.1 | 4.8 | 3.2 | 1.4 |
| | $720 \times 1440$ | 1.5 | 1.4 | 1.4 | 1.3 | 0.8 |
| | $1440 \times 2880$ | 0.4 | 0.4 | 0.4 | 0.4 | 0.3 |

Table 7: Average frames-per-second (fps) runtime for the PyTorch implemented ESM module on the ScanNet scene 0002-00, with RGB projections, running on 8 CPU cores.

| | | Monocular Res | | | | |
|---|---|---|---|---|---|---|
| | | $60 \times 80$ | $120 \times 160$ | $240 \times 320$ | $480 \times 640$ | $960 \times 1280$ |
| Memory Res | $45 \times 90$ | 108.2 | 105.5 | 92.5 | 86.1 | - |
| | $90 \times 180$ | 102.0 | 92.2 | 85.0 | 83.8 | - |
| | $180 \times 360$ | 81.8 | 79.9 | 76.2 | 72.0 | - |
| | $360 \times 720$ | 44.1 | 43.0 | 42.3 | 41.7 | - |
| | $720 \times 1440$ | 13.9 | 13.9 | 13.8 | 13.1 | - |
| | $1440 \times 2880$ | 3.7 | 3.7 | 3.7 | 3.6 | - |

Table 8: Average frames-per-second (fps) runtime for the PyTorch implemented ESM module on the ScanNet scene 0002-00, with RGB projections, running on Nvidia RTX 2080 GPU.

