# OpenReview forum: "End-to-End Egospheric Spatial Memory"
_ICLR.cc/2021/Conference — ICLR 2021 Poster_

### Official Review · AnonReviewer3 · 2020-10-28
**Ego-centric Spatial Memory Networks - Review**

**Rating:** 4
**Confidence:** 3

**Review:**

# Summary
-------
This paper presents a method to build an ego-centric spatial memory map from an agent's viewpoint. This map module is differentiable and can be used for a variety of tasks, such as object segmentation, or image-to-action learning in different control tasks.

# Pros
----
+ The idea of combining the ego-centric representation seems interesting and novel.
+ Evaluation in Image-to-Action learning shows good performance compared to baselines.

# Cons
----
   *  _[Major]_ The Method section is very confusing. The method is based on the EKF pipeline and modifies the particular parts of the pipeline. The authors only briefly outline the overview of the method and then jump into specific details.  The text mostly focuses on improvement, and hence it is hard to judge which parts are novel and what the contribution is. Furthermore, there are errors in notation and variables that are not properly explained. u_t is first defined as incremental pose measurements and then as a control vector. Kamera intrinsic matrix K_1 is never introduced. The method section would benefit from restructuring, especially giving a clear overview of the method would help the reader to understand the method better.

   * _[Major]_ Experiments are not well explained, and more baselines are necessary for proper evaluation. In all experiments, the experimental set-up is not well defined. For example, in the first experiment, we find out that the authors use imitation learning only from the appendix. In the second experiment, the environment used for the evaluation is not introduced. The experimental set-up needs to be clearly outlined, and the goal of the experiment needs to be provided. The choice of baselines is not well motivated. Why were LSTM and NTM chosen in the first experiment? There is very little rationale for the choice in the submission. Do these approaches show the state-of-the-art performance on the examined tasks? In the second experiment, the authors only use the LSTM network to train the RL agent. Various methods tackle the navigation problem in the RL domain, and hence the authors should choose better baselines for comparison. I suppose that the PO task is some form of navigation task, although the authors never explain what the PO task is.

   * _[Minor]_ The simple approach to directly quantize pixel projections leads to artifacts in the map. It is a bit unclear why this option was chosen, especially now that progress on differentiable renderers has been made.

---

> ### Author Response · Authors · 2020-11-13
> **Reviewer 3 Author Response (1/2)**
>
> We thank the reviewer for their helpful comments and feedback.
> We address the main points of concern below.
>
> **The authors only briefly outline the overview of the method and then jump into specific details. The method section would benefit from restructuring, especially giving a clear overview of the method would help the reader to understand the method better.**
>
> We thank the reviewer for pointing this out. We have modified the method section to introduce the module more clearly, and also extended the section to explain the integration of our ESM module in the network architectures used for the experiments. We hope this clarifies confusion, otherwise we are very happy to make further changes to benefit understanding.
>
> **The text mostly focuses on improvement, and hence it is hard to judge which parts are novel and what the contribution is.**
>
> We thank the reviewer for pointing this out. We have added a sentence in the introduction clearly outlining the key contribution: “To the best of our knowledge, ESM is the first end-to-end trainable egocentric memory with full panoramic representation, enabling direct encoding of the surrounding scene in a 2.5D image.”
>
> We hope this helps to clarify the central novelty of the method, but we are happy to make further changes to the method section if this remains unclear.
>
> **u_t is first defined as incremental pose measurements and then as a control vector.**
>
> We thank the reviewer for pointing this out. Our reasoning behind this was that control vectors in state estimation are usually measurements of some kind, for example, velocity or odometry measurements. Therefore, these two descriptions of u_t are not contradictory. However, to improve clarity, we now only use the term: “incremental pose measurement”.
>
> **Kamera intrinsic matrix K_1 is never introduced.**
>
> We thank the reviewer for pointing this out. The term K_1^(-1) was introduced as the inverse camera intrinsic matrix in the text just before Eq. 1. We assume the reviewer would prefer K_1 be introduced before K_1^(-1), and so we have now added a more explicit introduction to the K_1 parameter earlier in the same sentence.
>
> **Experiments are not well explained. In all experiments, the experimental set-up is not well defined.**
>
> We agree that the experiments were not clear in the original submission. The experiment descriptions have now been improved, and some important content has been moved from the appendices to the main body. We hope these changes help the reviewer to better understand the experiments - we are very happy to make further changes to benefit understanding.
>
> **The experimental set-up needs to be clearly outlined, and the goal of the experiment needs to be provided.**
>
> We have added a new paragraph at the beginning of the experiments section (Sec 4), which outlines the overall goal of the experiments. We have also improved the clarity of the experiments section. Quoting the new introductory paragraph at the beginning of section 4:
>
> “The goal of our experiments is to show the wide applicability of ESM to different embodied 3D learning tasks, where ESM outperforms existing memory baselines. We test two different applications:
>
> 1.Image-to-action learning for multi-DOF control (Sec 4.1).  Here we consider drone and robot manipulator target reacher tasks using either ego-centric or scene-centric cameras.We then assess the ability for ESMN policies to generalize between these different camera modalities, and assess the utility of the ESM geometry for obstacle avoidance.  We train policies both using imitation learning (IL) and reinforcement learning (RL).
>
> 2.Object segmentation (Sec 4.2). Here we explore the task of constructing a semantic map, and the effect of changing the ESM module location in the computation graph on performance.”
>
> **In the first experiment, we find out that the authors use imitation learning only from the appendix.**
>
> We thank the reviewer for pointing this out, this has now been moved to the main body.
>
> **In the second experiment, the environment used for the evaluation is not introduced.**
>
> We thank the reviewer for pointing this out. The description of the reinforcement learning experiment has now been extended, to properly introduce the environment. Quoting the revision: “We therefore train both ESMN-RGB and an LSTM baseline on a simpler variant of the MR task via DQN. We refer to this variant as MR-Seq, where the manipulator must reach red, blue and then yellow spherical targets from egocentric observations, after which the episode terminates. The only other difference to MR is that MR-Seq uses 128x128 images as opposed to 32x32.”
>
> We emphasize that the original MR task is modified from the RLBench reacher task, and further details about the task setup can be found in the code and paper for this benchmark.

---

> > ### Author Response · Authors · 2020-11-13
> > **Reviewer 3 Author Response (2/2)**
> >
> > **The choice of baselines is not well motivated. Why were LSTM and NTM chosen in the first experiment?
> > There is very little rationale for the choice in the submission. Do these approaches show the state-of-the-art performance on the examined tasks?**
> >
> > We have added a description at the beginning of section 4.1 clearly outlining the rationale behind our choice of baselines. Quoting this section: “While ego-centric cameras are typically used when learning to navigate planar scenes from images(Mnih et al., 2016; Jaderberg et al., 2016; Zhu et al., 2017; Gupta et al., 2017; Parisotto & Salakhut-dinov, 2017), static scene-centric cameras are the de facto when learning multi-DOF controllers for robot manipulators (Levine et al., 2016; James et al., 2017; Matas et al., 2018; James et al.,2019b). We consider the more challenging and less explored setup of learning multi-DOF visuomotor controllers from ego-centric cameras, and also from moving scene-centric cameras. LSTMs are the de facto memory architecture in the RL literature (Mnih et al., 2016; Jaderberg et al., 2016; Espeholtet al., 2018; Kapturowski et al., 2018), making this a suitable baseline.  NTMs represent another suitable baseline, which have outperformed LSTMs on visual navigation tasks (Wayne et al., 2018). Many other works exist which outperform LSTMs for planar navigation in 2D maze-like environments(Gupta et al., 2017; Parisotto & Salakhutdinov, 2017; Henriques & Vedaldi, 2018), but the top-down representation means these methods are not readily applicable to our multi-DOF control tasks. LSTM and NTM are therefore selected as competitive baselines for comparison.”
> >
> > **In the second experiment, the authors only use the LSTM network to train the RL agent. Various methods tackle the navigation problem in the RL domain, and hence the authors should choose better baselines for comparison.**
> >
> > Referring to the beginning of section 4.1, and also to our earlier response on the choice of baselines, we articulate why we find the LSTM and NTM methods to be strong baselines for the tasks we test on. Quoting section 4.1: "LSTMs are the de facto memory architecture in the RL literature (Mnih et al., 2016; Jaderberg et al., 2016; Espeholtet al., 2018; Kapturowski et al., 2018), making this a suitable baseline.  NTMs represent another suitable baseline, which have outperformed LSTMs on visual navigation tasks (Wayne et al., 2018)."
> >
> > We do however agree that some other baselines could further strengthen our results. Auxiliary losses are a common addition to improve LSTM learning (Jaderberg et al., Reinforcement learning with unsupervised auxiliary tasks, 2016,  Sadeghi et al., Sim2Real View Invariant Visual Servoing by Recurrent Control, 2017,  Mirowski et al., Learning to Navigate in Cities Without a Map, 2019), and so we are running experiments which combine the LSTM baseline with the auxiliary losses as used in (Sadeghi et al., 2017, Mirowski et al., 2019). We feel this baseline plus the existing baselines represent a comprehensive set of competitive baselines for the tasks we evaluate on. We don’t believe this additional baseline will change the results or conclusions of the paper. If the reviewer has specific examples of additional baselines they feel are missing, we would be very happy to run additional experiments to further validate our results.
> >
> > **I suppose that the PO task is some form of navigation task, although the authors never explain what the PO task is.**
> >
> > In the original submission, PO in this context referred to a baseline which we called Partial-Omni. It did not stand for partial observability as introduced in the abstract. To avoid this clash of terminology, we now call this baseline Partial-Omni-Oracle (PO2). We have also moved the explanation of PO2 from the appendices into the main text, in section 4.1.
> >
> > To further explain, PO2 is a baseline which uses a ground-truth omni-directional camera of the same dimensions as ESM (90x180), but with ESM re-projections used to mask the observed pixels throughout the image sequence. Quoting the revised paper: “PO2 cannot see regions where the monocular camera has not looked, but it maintains a pixel-perfect memory of anywhere it has looked.”
> >
> > **The simple approach to directly quantize pixel projections leads to artifacts in the map. It is a bit unclear why this option was chosen, especially now that progress on differentiable renderers has been made.**
> >
> > This was chosen for simplicity. Our module is implemented using core tensor operations alone, with scatter_nd responsible for the projections and depth buffer. Scatter functions are available in all major deep learning frameworks, whereas differentiable rendering functions are not. We see benefit in the implementational simplicity, particularly when our results indicate that this simpler method is sufficient. Having said this, the ESM module could easily be modified to use rasterization and mesh rendering, without change to other aspects of the method.

---

### Official Review · AnonReviewer2 · 2020-10-28
**Interesting approach to egocentric memory based on forward warping; experiments cover a wide range of tasks, but are very hard to understand as there is little / no explanation of the evaluated methods; some claims seem too strong**

**Rating:** 7
**Confidence:** 4

**Review:**

The paper considers the problem of creating spatial memory representations, which play important roles in robotics and are crucial for real-world applications of intelligent agents. The paper proposes an ego-centric representation that stores depth values and features at each pixel in a panorama. Given the relative pose between frames, the representation from the previous frame is transformed via forward warping (using known depth values) to the viewpoint of the current frame. The proposed approach has no learnable parameters. Experiments on a wide range of tasks show that the proposed approach outperforms baselines such as LSTM and NTM.

On the positive side, the approach is positively simple, in the sense that it relies on known techniques (EKF, forward warping, etc.) that ensure that it is easy to implement while achieving good results in the experiments. Up to Sec. 4, I found the paper easy to follow, although some design choices could be better motivated (e.g., I assume that diagonal covariances are assumed for simplicity).
The paper evaluates the proposed approach on multiple tasks and in various configurations, which is another strength of the paper.

While I like the proposed approach, I also see multiple significant weaknesses:
1) I found the experimental evaluation nearly impossible to understand. My main problem is that I don't understand what the different method that are evaluated are:
 * Given the abbreviation ESMN introduced in the abstract, I assume that ESMN is the proposed approach. ESM seems to be a variant of ESMN, but I am not sure how ESM and ESMN differ as the difference is never clearly described (or if it is, I seem to have missed it). Sec. 4.1.1 mentions training with a convolutional encoder in the context of ESMN, Sec. 4.1.3 and Sec. 4.1.4 only evaluate ESM but not ESMN, while Sec. 4.2 states that "ESM represents map-only inference, while ESMN includes convolutions for both the image-level and map-level inference". Unfortunately, the term "map-level" inference is not well-defined. Overall, I don't understand the difference between ESM and ESMN. As a result, it is unclear to me why ESM performs worse than ESMN in Tab. 1 for DR-Ego-S but comparable for all other tasks in the table, or why ESM and not ESMN is used for some of the experiments. Similarly, what is the "ESM-DepthAvoid" baseline? Does it only use depth and no features?
 * Tab. 1 contains a baseline called "PO" and I don't understand how it works. The abstract introduces PO as an abbreviation for partial observability, but that does not seem to be an explanation for a baseline. The PO baseline performs similarly well as ESM and ESMN in Tab. 1, which makes me wonder whether Tab. 1 really shows the superiority of the proposed approach.
 * I am confused by the statement "In contrast, ESM by design stores features in the memory with meaningful indexing. The inclusion of relative cartesian co-ordinates in the memory image also effectively aligns each pixel with an associated relative translation." since the inclusion of such coordinates is never mentioned before. I assume ESM uses some form of handcrafted features?
2) Some of the statements made in the paper seem too strong:
 * I don't see how the claim "Our memory is much more expressive than these 2D examples, with the ability to represent detailed 3D geometry in all directions around the agent." (Sec. 2.2)  holds. I agree that being able to store information in 2.5D (panorama and depth) is more powerful than storing only top-down 2D maps. However, the allocentric maps of the references can store larger and more complicated scene parts. E.g., if an agent would turn around a corner, ESMN would essentially be forced to forget about everything that is not directly visible anymore as it is occluded and thus not included in the memory structure anymore. As such, a point can be made that ESMN is much less expressive than for example Henriques & Vedaldi. Given that the latter evaluate on significantly more complex scenes compared to the ones used in this paper (which do not have strong occlusions) strengthens this impression.
 * Regarding the statement "Although the most recent depth frame is also a strong signal for local obstacle avoidance, we show that the avoidance based on the full ESM geometry results in fewer collisions when tested on a variant of the drone task with the inclusion of 25 obstacles.": Looking at Tab. 2, it seems to me that the standard deviation is so large for both ESM and ESM-DepthAvoid that it is unclear to me whether one is consistently better than the other.
3) The CodeSLAM approach from Bloesch et al. also provides a form of memory representation and I don't understand why the paper, and its follow-up (Zhi et al., SceneCode: Monocular Dense Semantic Reconstruction using Learned Encoded
Scene Representations, CVPR 2019), is not discussed in the related work section.

Overall, I believe that the paper has potential. My main criticism is that I do not feel able to properly understand the experimental evaluation. As such, it is hard to recommend acceptance. However, I am willing to increase my score if the information necessary to understand this part of the paper is provided.

### After rebuttal phase ###
The answers provided by the authors and the revised version of the paper sufficiently address my concerns. As such, I recommend to accept the paper. I am still concerned that the experimental evaluation is very packed with multiple experiments while lacking details on the experimental setup and explanations of the baselines. Still, I feel that in the current form, the paper can be accepted.

---

> ### Author Response · Authors · 2020-11-13
> **Reviewer 2 Author Response (1/2)**
>
> We thank the reviewer for their helpful comments and feedback.
> We address the main points of concern below.
>
> **I am not sure how ESM and ESMN differ as the difference is never clearly described**
>
> We acknowledge that this was unclear in the original submission, and have now added network diagrams clearly outlining the differences in Figure 2. We have also slightly changed terminology to avoid such confusion. Before, we used ESM interchangeably to refer to both the standalone module and to a network using the module without convolutions beforehand, with RGB values projected into the module.
>
> We now refer to ESM purely as the standalone parameter-free memory module, and refer to the two network variants which make use of this module as ESMN-RGB and ESMN, to keep the distinctions clear.
>
> In terms of the difference between ESMN and ESMN-RGB, ESMN projects convolutional features into the ESM module, whereas ESMN-RGB directly projects RGB features from the acquired images into the ESM module. Again, Figure 2 outlines this architectural difference more clearly.
>
> **What is the "ESM-DepthAvoid" baseline? Does it only use depth and no features?**
>
> We have renamed this baseline as “Single Depth Frame Avoidance” for better clarity, and added an explanation in the obstacle avoidance section.
>
> To answer your question directly, this baseline performs obstacle avoidance using only the geometry available from the most recent depth frame. In contrast, “ESM Depth Map Avoidance” uses the depth information stored in the full ESM memory.
>
> **Tab. 1 contains a baseline called "PO" and I don't understand how it works.**
>
> In the original submission, PO in this context referred to a baseline which we called Partial-Omni. It did not stand for partial observability as introduced in the abstract. To avoid this clash of terminology, we now call this baseline Partial-Omni-Oracle (PO2). We have also moved the explanation of PO2 from the appendices into the main text, in section 4.1.1.
>
> PO2 is a baseline which uses a ground-truth omni-directional camera of the same dimensions as ESM (90x180), but with ESM re-projections used to mask the observed pixels throughout the image sequence. Quoting the revised paper: “PO2 cannot see regions where the monocular camera has not looked, but it maintains a pixel-perfect memory of anywhere it has looked.”
>
> **I am confused by the statement "In contrast, ESM by design stores features in the memory with meaningful indexing. The inclusion of relative cartesian coordinates in the memory image also effectively aligns each pixel with an associated relative translation." since the inclusion of such coordinates is never mentioned before. I assume ESM uses some form of handcrafted features?**
>
> This statement in the original submission was actually intended to say “the inclusion of polar coordinates in memory”. We apologise for this mistake.
>
> We have modified the method section to bring earlier attention to these coordinates. We now consider the polar coordinates as part of the egosphere state. Quoting the beginning of our revised section 3: “The egosphere image consists of 2 channels for the polar and azimuthal angles, 1 for radial depth, and n for encoded features. The angles are not included in the covariance, as their values are implicit in the egosphere image pixel indices. The covariance only represents the uncertainty in depth and features at these fixed equidistant indices.”
>
> We then refer to these again in the revised section 3.3: “The inclusion of polar angles, azimuthal angles and depth means the full relative polar coordinates are explicitly represented for each pixel in memory”
>
> Finally, quoting our revision of the sentence which caused confusion: “ESM by design stores the encoded features in memory with meaningful indexing. The ESM structure ensures that the encoded features for each pixel are aligned with the associated relative polar translation, represented as an additional feature in memory.”
>
> We hope that these revisions clarify the original confusion.
>
> **I don't see how the claim "Our memory is much more expressive than these 2D examples, with the ability to represent detailed 3D geometry in all directions around the agent." (Sec. 2.2) holds.**
>
> We agree that this statement was too simplistic and strong, and we have modified the statement with a more balanced comparison. Indeed, the benefits of our approach (detailed immediate unoccluded local geometry with orientation aware indexing) are complementary to 2D methods (occlusion aware planar understanding for navigation).
>
>
> The sentence now reads: “Our memory instead focuses on local perception, with the ability to represent detailed 3D geometry in all directions around the agent. The benefits of our module are complementary to existing 2D methods, which instead focus on occlusion aware planar understanding suitable for navigation.”

---

> > ### Author Response · Authors · 2020-11-13
> > **Reviewer 2 Author Response (2/2)**
> >
> > **Looking at Tab. 2, it seems to me that the standard deviation is so large for both ESM and EMM-DepthAvoid that it is unclear to me whether one is consistently better than the other.**
> >
> > In the original results, we ran the avoidance controller at the same frequency as the ESM updates, which resulted in a large number of collisions even when the obstacles were well detected. We now run the avoidance algorithm at 10x the ESM rate, and observe much clearer distinctions between the baselines. “ESM Depth Map Avoidance” now outperforms “Single Depth Frame Avoidance” by more than one standard deviation in both columns 4 and 6 of Table 2.
> >
> > **The CodeSLAM approach from Bloesch et al. also provides a form of memory representation and I don't understand why the paper, and its follow-up (Zhi et al., SceneCode: Monocular Dense Semantic Reconstruction using Learned Encoded Scene Representations, CVPR 2019), is not discussed in the related work section.**
> >
> > The paper is mentioned in the related work section on the original upload, but was not significantly discussed. Our reasoning for this was that CodeSLAM does not use a fixed-size memory to encode the surrounding geometry. Rather, multiple keyframes are used, each of which optimizes a keyframe-specific code. The scene geometry is then composed of the individual keyframes projected into world space.
> >
> > However, we do acknowledge that this distinction could be made more clearly, and so in the revision we have extended our discussion of CodeSLAM in the related work section. The new sentence reads: “Taking CodeSLAM as an example, keyframe removal strategies are needed to prevent the map size from growing indefinitely.”
> >
> > We also agree that SceneCode is a clear omission, particularly in light of our object segmentation experiments. We have added a short discussion of this in the object segmentation experiment section. Quoting the latest revision: “One approach is to perform image-level segmentation only in individual monocular frames, and then perform probabilistic fusion when projecting into the map… Another approach is to first construct an RGB map, and then pass this map as input to a network… SceneCode takes a different approach, and combines monocular predictions with multi-view optimization to gain the benefits of wider surrounding context.”
> >
> > **Some design choices could be better motivated (e.g., I assume that diagonal covariances are assumed for simplicity).**
> >
> > We thank the reviewer for pointing this out. We agree this could be made more clear, and we will include a sentence in the method section for the next paper revision stating: "Diagonal covariances are assumed due to the large state size of the images, for which the full covariance matrices cannot be efficiently stored in memory."
> >
> > Elaborating on this, the egosphere state in our experiments is 90x180. The covariance is then represented by an image of 16200 variance values using the diagonal assumption. The full covariance would include ~262 million values, which would dominate GPU memory.

---

> > > ### Comment · AnonReviewer2 · 2020-11-24
> > > **The revised version addresses my concerns**
> > >
> > > Thank you very much for the detailed answers to my questions and the revised version of the paper.
> > > Your answers and the revision address my concerns and I willing to change my rating to recommend acceptance.
> > >
> > > I think that the statement “Taking CodeSLAM as an example, keyframe removal strategies are needed to prevent the map size from growing indefinitely.” is a bit unfair towards CodeSLAM. ESM can only remember parts of the scene that are still visible. While CodeSLAM has the potential to "remember" larger areas, one could simply choose to forget old observations by dropping all but the last k keyframes. This strategy is commonly employed by visual odometry approaches (which in contrast to SLAM only keep local maps). As such, one does not need to develop keyframe removal strategies as suggested by the text.
> > > Overall, I am not convinced that egocentric representations are better than allocentric maps. In my opinion, the latter can easily encompass the former (by simply forgetting older observations) while the former cannot model the latter. On the downside, allocentric maps are harder to construct and maintain. As such, I see the paper as an interesting step towards comparing both types of representations. I see ESMs as an alternative to more complex maps and determining in which scenarios they are sufficient is an interesting question.
> > >
> > > In my personal opinion, I still think that the experimental part of the paper is a bit hard to read. There are a lot of experiments and many baselines, but few of them are really described in detail. Many of these details are provided in the appendices, but this makes the paper hard to read as one needs to jump back and forth. Using a subset of the experiments, with a more detailed description of the experiments and the baselines, and providing the rest in the appendices would, in my opinion, improve readability. But this is my personal opinion and definitively not a reason for rejection.

---

> > > > ### Author Response · Authors · 2020-11-25
> > > > **Reviewer 2 Author Response**
> > > >
> > > > Thank you for responding to our comments; we appreciate you taking the time to give further feedback, and we are very happy to hear that our answers and revisions have addressed most of your concerns.
> > > >
> > > > We take on board the comment about the "keyframe removal strategies" being slightly unfair to CodeSLAM, and we also acknowledge your general thoughts on allocentric vs egocentric formulations, and the density of the experiment section. We respond to these points below.
> > > >
> > > > Firstly, we acknowledge that many simple and effective heuristics exist for keyframe selection, which combine time and space constraints, and we agree that preventing map growth should not be the only motivation for the egocentric formulation over allocentric approaches.
> > > >
> > > > In the latest revision, we have removed the sentence “Taking CodeSLAM as an example, keyframe removal strategies are needed to prevent the map size from growing indefinitely.”. We have added a sentence elaborating on the benefits of the structured memory and egocentric indexing for downstream tasks. Quoting the latest revision: “... Unlike allocentric formulations, the memory indexing is fully coupled to the agent pose, resulting in an ordered representation particularly well suited for downstream egocentric tasks, such as action selection.”
> > > >
> > > > Elaborating further on this general point, it is not clear how keyframe-based methods (like CodeSLAM) can represent the memory in a compact and ordered fashion suitable for conditioning a downstream reactive policy network. The keyframes could be stacked channelwise for example, but frame-stacking is already well explored in image-to-action learning, and even with sensible frame-skipping to maximise scene visibility, this is generally inferior to the use of dedicated memory such as LSTMs. Finding a better way of conditioning a policy on the keyframes is non-trivial. ESM instead represents the memory compactly as a single egocentric image, which can be input to CNNs directly, and outperforms existing memory baselines.
> > > >
> > > > Regarding the general egocentric vs. allocentric debate, we agree that egocentric mapping is by no means a replacement for allocentric mapping, and indeed, allocentric is inherently more expressive than egocentric. As you pointed out, our work explores contexts where local spatial memory is sufficient, and specifically how the egocentric formulation enables direct integration into neural networks for learning tasks in an end-to-end manner.
> > > >
> > > > Finally, regarding the experiment section, we acknowledge that there is a lot of content, and that the descriptions are quite compact as a result. We will explore moving some experiments to the Appendix and expanding the explanations in time for the camera-ready paper if accepted.
> > > >
> > > > Again, we thank the reviewer for their very helpful comments, which have greatly helped to shape the paper. We look forward to Jan 12th.

---

### Official Review · AnonReviewer1 · 2020-10-29
**Reivew**

**Rating:** 6
**Confidence:** 3

**Review:**

The paper introduced Egocentric Spatial Memory Networks (ESMN), a novel learning paradigm and architecture for encoding spatial memory in a sphere representation. The representation can be used for several downstream applications ranging from semantic segmentation as well as action learning for robotics applications.

The formulation of the proposed approach builds on a Kalman Filter setting and encodes memory in a spherical structure. Both the formulation and design makes sense, and the downstream applications show that the proposed approach is indeed plausible.

While I think the value of this paper is well justified, I do have a few comments on the paper:
- I think it'll be worthwhile to elaborate on the "Mono" method, including the network architectures as well as training details and such. It will also help understand which part of ESMN improves over this baseline method.
- In the related work section, one argument made by authors is that ESMN and MemNN/NTM are two different paradigms for learning spatial memory. While the proposed approach ESMN is reasonable, I wonder if it is possible to have a concrete experiment comparing those two different designs.
- Could the authors elaborate on Sec 4.1.1 Fig 5? What's the meaning of the colorings in this figure, and how to interpret that?

---

> ### Author Response · Authors · 2020-11-13
> **Reviewer 1 Author Response**
>
> We thank the reviewer for their helpful comments and feedback.
> We address the main points of concern below.
>
> **I think it'll be worthwhile to elaborate on the "Mono" method, including the network architectures as well as training details and such. It will also help understand which part of ESMN improves over this baseline method.**
>
> We agree, and we have added a high-level schematic of the architectural differences between Mono, LSTM/NTM, ESMN-RGB , and ESMN in Figure 2. We have also provided the full network architectures for the imitation learning and object segmentation experiments in the Appendices.
>
> **In the related work section, one argument made by authors is that ESMN and MemNN/NTM are two different paradigms for learning spatial memory. While the proposed approach ESMN is reasonable, I wonder if it is possible to have a concrete experiment comparing those two different designs.**
>
> We do compare ESMN against NTM in our imitation learning experiments, the results are presented in Table. 1. For improved performance, we implement the same changes proposed by (Wayne et al., Unsupervised Predictive Memory in a Goal-Directed Agent, 2018). These changes are fully explained in the Appendices.
>
> **Could the authors elaborate on Sec 4.1.1 Fig 5? What's the meaning of the colorings in this figure, and how to interpret that?**
>
> We thank the reviewer for pointing this out. We have improved the explanations of this figure both in the main text and in the figure caption. To answer the question directly, these images show the 2D representation of the features stored in the spherical ESM memory (in the same way that an atlas shows a 2D representation of a spherical globe), for adjacent timesteps in different reacher tasks, going chronologically from left to right. The rows correspond to separate chronological image sequences.

---

### Author Response · Authors · 2020-11-13
**Author Comments on the Paper Revision**

We thank all reviewers for their helpful comments and feedback.

We have uploaded a new revision of the paper, and responded directly to each reviewer, outlining where our revised paper addresses the reviewer’s comments.

We provide a brief summary of the major changes as a result of the reviewers’ comments below:

1. Clearer explanations and diagrams outlining the method, in particular Figure 1.

2. Diagram of the different network architectures, clearly outlining the difference between ESMN and ESMN-RGB, as well as the Mono and LSTM/NTM baselines in Figure 2.

3. Improved explanation for the choice of experiments and motivation, improved explanations of the experiments, and better justification for the choice of baselines in section 4.

4. Improved clarity for Figure 4, which presents sequences of the full RGB memory through time.

Other changes have been made which are not in direct response to reviewer comments. We list these additional changes here for the reference of all reviewers:

1. Paper title changed from “Ego-Centric Spatial Memory Networks” to “End-to-End Egospheric Spatial Memory”, in order to better capture the novelty of the method in the spherical representation.

2. Added new experimental results on reacher tasks conditioned on shape ID, further showing the benefits of learning features through the memory, using the ESMN architecture.

3. Added new experimental results for policies generalizing between ego-centric and scene-centric image acquisition, and added a diagram showing example image trajectories for both modes of acquisition.

4. Added new experimental results for the obstacle avoidance task, following improvements to the local avoidance algorithm. The benefits of the ESM geometry over single depth frame avoidance is now more evident in the results.

5. Added new experimental results for the object segmentation task, which explore the effects of changing monocular image resolution, memory resolution, and image sequence length on segmentation accuracy. Also added an improved diagram showing the object segmentation predictions, and failure modes of the mono method which ESMN is able to overcome.

In addition to these changes to the paper, we have also added a 5th video to the shared site showing object segmentation results on a test scene from the ScanNet dataset. The shared site can be found at:
https://sites.google.com/view/egocentric-spatial-memory-nets

We hope these changes make the paper easier to follow, and the inclusion of new results create a stronger impression of the utility of our ESM module in different learning contexts involving spatial reasoning.

---

### Author Response · Authors · 2020-11-17
**Author Comments on the 2nd Paper Revision**

To further address some of the reviewer comments initially raised, we have uploaded a 2nd revision to the paper.
We provide a brief summary of the main changes below:

1. We extend the Appendix to include the training and validation curves for each of the imitation learning networks, as well as a discussion on these curves in light of the policy performances. We also more thoroughly outline the finer details for each of the experiments throughout the paper.

2. Following on from the request for additional baselines from Reviewer 3, we added a new baseline (LSTM-Aux) which combines auxiliary losses adapted from two papers which propose LSTM-based image-to-action learning pipelines. We include further details of this auxiliary baseline in the Appendix.

3. Other smaller fixes/improvements to some figures and explanations throughout the text.

---

### Author Response · Authors · 2020-11-25
**Author Comments on the 3rd Paper Revision**

To further address some of the reviewer comments raised, we have uploaded a 3rd and final revision to the paper with regards to the discussion phase. We would like to again thank all reviewers for taking the time to provide very helpful feedback. We hope our answers to your questions and our revisions adequately address the concerns raised.

We provide a brief summary of the main changes in the latest revision below:

1. The RL experiments are extended to include all 7 baseline methods, the same as used in the imitation learning experiments. This means seven methods are now evaluated on the sequential reacher task trained via RL, as opposed to only two in the previous revision. We hope this addresses concerns raised by Reviewer 3 regarding lack of RL baselines.

2. A discussion on the LSTM auxiliary basline has been added to the appendices, with the losses on the validation set during imitation learning included in new plots. We outline overfitting as a failure mode of the auxiliary baselines, and expand discussions on the difficulty in tuning auxiliary losses for different tasks. We hope these plots and discussion further address concerns raised by Reviewer 3 regarding the need for additional baselines.

3. Principal component visualizations of the implicit features stored in memory by the pre-ESM encoder (from the ESMN architecture) have been added to the appendices, as well as discussions on the meaning of these. This provides further insights into the manner in which ESMN succeeds at the different reacher tasks where ESMN-RGB fails. The encoded features visibly represent concepts such as foreground, background, and object edges, which are more expressive than color values alone.

4. A runtime analysis has been included in the appendices. We compare across deep learning framework, computing device, and monocular and memory image dimensions. ESM with medium sized images, for example 480x640 monocular images and 360x720 memory images, is able to run at ~60fps. These results help to reinforce the statement in the abstract that: "the module forms a bridge between real-time mapping systems and differentiable memory architectures".

---

### Decision · Program_Chairs · 2021-01-07
**Final Decision**

**Decision:**

Accept (Poster)

**Comment:**

In this paper, the authors combine ideas from SLAM (using an Extended Kalman Filter and a state with nonlinear transitions and warping) and differentiable memory networks that store a spherical representation of the state (from the ego-centric point of view of an RL agent moving in an environment) with depth and visual features stored at each pixel and dynamics transitions corresponding to warping.

The main idea in the paper is very simple and elegant, but I will concur with the reviewers that the writing of the first version of the paper was extremely hard to understand and that the experimental section was too dense. Two subsequent revisions of the paper have dramatically improved the paper.

Given the spread of scores (R1: 6, R2: 7 and R3: 4) and the fact that only R1 and R2 have acknowledged the revisions, I will veer towards acceptance.